# Wood Waste from Fruit Trees: Biomolecules and Their Applications in Agri-Food Industry

**DOI:** 10.3390/biom12020238

**Published:** 2022-02-01

**Authors:** Maria Jose Aliaño-González, Julien Gabaston, Victor Ortiz-Somovilla, Emma Cantos-Villar

**Affiliations:** 1IFAPA Rancho de la Merced, Consejería de Agricultura, Ganadería, Pesca y Desarrollo Sostenible, Junta de Andalucía, 11471 Jerez de la Frontera, Spain; mariaj.aliano@juntadeandalucia.es; 2Departamento de Química Analítica, Facultad de Ciencias, Universidad de Cádiz, 11510 Cadiz, Spain; 3Departamento de Nutrición y Bromatología, Toxicología y Medicina Legal, Área de Nutrición y Bromatología, Facultad de Farmacia, Universidad de Sevilla, 41012 Seville, Spain; julien.gabaston@gmail.com; 4IFAPA Alameda del Obispo, Consejería de Agricultura, Ganadería, Pesca y Desarrollo Sostenible, Junta de Andalucía, Avenida Menéndez Pidal, 14004 Córdoba, Spain; victor.ortiz@juntadeandalucia.es

**Keywords:** wood waste, olive trees, grapevine, nut trees, stone fruit trees, pome trees, citrus fruit trees, bioactive compounds, phenolic compounds, circular economy

## Abstract

In the European Union (EU), a total of 11,301,345 hectares are dedicated to the cultivation of fruit trees, mainly olive orchards, grapevines, nut trees (almond, walnut, chestnut, hazelnut, and pistachio), apple and pear trees, stone fruit trees (peach, nectarine, apricot, cherry, and plum), and citrus fruit trees (orange, clementine, satsuma, mandarin, lemon, grapefruit, and pomelo). Pruning these trees, together with plantation removal to a lesser extent, produces a huge amount of wood waste. A theoretical calculation of the wood waste in the European Union estimates approximately 2 and 25 million tons from wood plantation removal and pruning, respectively, per year. This wood waste is usually destroyed by in-field burning or crushing into the soil, which result in no direct economic benefits. However, wood from tree pruning, which is enriched in high added-value molecules, offers a wide spectrum of possibilities for its valorization. This review focuses on the contribution of wood waste to both sustainability and the circular economy, considering its use not only as biomass but also as a potential source of bioactive compounds. The main bioactive compounds found in wood are polyphenols, terpenes, polysaccharides, organic compounds, fatty acids, and alkaloids. Polyphenols are the most ubiquitous compounds in wood. Large amounts of hydroxytyrosol (up to 25 g/kg dw), resveratrol (up to 66 g/kg dw), protocatechuic acid (up to 16.4 g/kg), and proanthocyanins (8.5 g/kg dw) have been found in the wood from olive trees, grapevines, almond trees and plum trees, respectively. The bioactivity of these compounds has been demonstrated at lower concentrations, mainly in vitro studies. Bioactive compounds present antioxidant, antimicrobial, antifungal, biostimulant, anti-inflammatory, cardioprotective, and anticarcinogenic properties, among others. Therefore, wood extracts might have several applications in agriculture, medicine, and the food, pharmaceutical, nutraceutical, and cosmetics industries. For example, olive tree wood extract reduced thrombin-induced platelet aggregation in vitro; grapevine tree wood extract acts a preservative in wine, replacing SO_2_; chestnut tree wood extract has antifungal properties on postharvest pathogens in vitro; and stone tree wood extracts are used for aging both wines and brandies. Moreover, the use of wood waste contributes to the move towards both a more sustainable development and a circular economy.

## 1. The Abundance and Importance of Fruit Trees in Europe

In the EU, a total of 11,301,345 hectares (ha) are dedicated to the cultivation of fruit trees (Figure 1A). Olive orchards cover the largest area, with more than 5 million hectares (45% of the total area), followed by vineyards, with more than 3 million ha (28%). Almond trees and other nuts are the third most abundant (11%), with more than 1,200,000 ha. Meanwhile, 611,540 ha (5%) are dedicated to apple and pears trees 609,710 ha (5%) are dedicated to stone fruits (peaches, nectarines, apricots, cherries, and plums), with citrus fruits covering 502,366 ha (4%). The last group of fruit crops includes figs, avocados, kiwis, other tropical fruits, and bananas, which cover 141,000 ha (around 1.3%).

Regarding the distribution of the fruit growing area in the EU member countries, Spain stands out with 43% of the total surface, followed by Italy with 21%, Greece with 10%, France with 8%, and Portugal with 6% (Figure 1B).

Olives and grapes are the two major perennial crop systems traditionally grown in the Mediterranean Basin. Regarding **olive production**, Spain is by far the largest producer, responsible for over 50% of EU production, followed by Italy, Greece, and Portugal (Figure 2A). Other leading olive-producing countries by annual metric tons are France and Croatia. 

**Grapes** are an important cultural, economic, and ecological feature of the Mediterranean Basin but also a cosmopolitan crop, with the largest acreage and the highest economic value among fruit crops globally. The EU members with the highest grapevine surfaces are Spain, France, and Italy (Figure 2A).

The production of edible **nuts** in Europe has gradually increased since 2013. The most-produced edible nuts in Europe are hazelnuts and almonds, while walnuts, chestnuts, and pistachio nuts are produced to a lesser extent. Spain, the Netherlands, and Italy are major producers of processed edible nuts. In terms of primary production (growing, harvesting, and drying), the largest European producers are Spain (led by almonds), Italy (hazelnuts), Portugal (chestnuts), and France (walnuts) (Figure 2B). 

Thousands of varieties of **apples** are grown worldwide, many of which have been created and selected to grow in varied climates. This has enabled commercial apple production to take place in almost all member states. Just over a quarter (26.6%) of the EU-27′s harvested apple production came from Poland in 2019. The other principal apple-producing member states were Italy (19.9%) and France (15.1%) (Figure 2B). 

By contrast, the production of oranges and peaches is much more restricted by climatic conditions, and about 93% of these crops produced in the EU-27 come from Spain, Italy, and Greece (Figure 2B).

The information presented above means that Spain can be considered an important fruit producer. Moreover, apart from the fruits, other aspects may be considered. The total area of fruit tree cultivation in the EU is 11.33 Mha, thus making the EU an important producer of wood waste from fruit trees, due to their annual pruning. A theoretical calculation of the potential pruning availability of the EU estimates approximately 25 million tons (MT) per year of wood dry weight [2]. More specifically, 8 MT of wood waste is produced in Spain every year.

When considering a fruit tree plantation managed according to sustainability criteria, a wide range of ecosystem services can be detected. Indeed, apart from fruit production, these services may include climate change mitigation by reducing greenhouse gas (GHG) emissions, fossil fuel consumption and increasing carbon sequestration, soil nutrient recovery and cycling, improving water use efficiency and water regulation, preventing run-off and controlling erosion, the biological control of pests and diseases, biodiversity, pollination, and many others [2].

The following section focuses on the contribution of wood waste to both sustainability and the circular economy, considering its use not only as biomass but also as a potential source of bioactive compounds.

## 2. The Contribution of Wood Waste to a Circular Economy and Sustainable Bioeconomy

According to the United Nations, the global population is expected to increase from 7.7 billion (2019) to 9.7 billion in 2050 [3]. This prospect raises several concerns about the global consumption of biomass, fossil fuels, metals, and minerals, which is likely to double, and annual waste production, following the current trend, will increase by 70% in the next 40 years [4]. Undoubtedly, these premises challenge the move toward both a more sustainable development and a circular economy.

The circular economy is a model of production and consumption that involves sharing, leasing, reusing, repairing, refurbishing, and recycling existing materials and products as much as possible, thereby extending the life cycle of products. Within an envisioned circular bio-based economy, a key component is the valorization of biomass wastes and residues into valuable products. In practice, it implies reducing waste to a minimum. When a product reaches the end of its life, its materials are kept within the economy wherever possible. These can be productively used again and again, thereby creating further value [5]. The EU is set to transition from a linear to a circular economy, turning waste into a resource in order to increase resource efficiency and close the loop in a circular economy [6].

Wood comes mainly from forest biomass [7]. The main wood-based products are: (i) cork and wood, (ii) pulp and waste paper, (iii) cork and wood manufacturers (excluding furniture), and (iv) paper, paperboard, and by-products [8]. However, in other cases, wood is not produced biomass but, rather, residual biomass, which is the case of wood from the pruning of fruit trees.

Fruit tree pruning is an important process in the management of many orchards. Its main purpose is to facilitate crown shaping for optimal fruit production and efficient harvesting. A properly formed crown is able to produce and bear a large fruit yield [9]. Pruning is performed at least once a year to promote continual and optimal production through the control of tree physiology. In fact, by cutting tree shoots, the number of fruits per plant is reduced, thus allowing a better distribution of nutrients throughout trees, an effective canopy light exposition, and an optimal bearing potential of re-growth, therefore achieving quantitative and qualitative improvements in fruit crops [10].

Great differences in wood production can be found between fruit trees and even in the same fruit tree. Wood waste production depends on factors, such as the type of crop, variety and age, the form of the tree, density, the type of pruning, the climate and soil conditions, and other agronomic operations, such as irrigation [11,12]. On average, annual pruning from crops in good climatic and agronomic conditions can usually produce from 0.5 to 2.0 tons/ha of wood (dry weight, dw). 

Considering the current fruit tree area in Europe (11.33 Mha), around 25 MT of wood dw are currently produced each year. This wood must be removed for both disease control and to facilitate future tending activities. Consequently, it is usually destroyed by in-field burning or by crushing onto the soil, which results in no direct economic benefits [12]. Collecting the wood from pruning poses a series of logistical difficulties due to several factors: (i) its dispersion in the territory, (ii) the size and layout of the plantations, and (iii) the production of biomass per hectare is low in comparison to forestry wood [13].

However, this waste has been proposed as an excellent biomass source. Biomass from pruning, being a residual material, does not create any additional demand for land and can deliver substantial GHG emission savings compared to fossil fuels, reducing both the pressure on fossil fuel reserves and energy dependence. Pruning residues, when removed from the cultivated field, can be directed to energy conversion to obtain renewable forms of energy (heat and power), thus offering a supplemental “provisioning” service. This option is of relevance, considering that replacing the use of fossil fuels with renewables is a relevant international goal, within the frame of the former Kyoto protocol (from 2005 to 2020) and the Paris agreement thereafter (from 2020 to 2030 and 2050).

With the use of agroforest waste as biomass, fossil fuels are being displaced by renewable energy sources, and, consequently, the concentration of carbon dioxide is gradually reduced from the atmosphere, due to the “carbon neutrality” of biomass feedstock, thus mitigating climate change [2,14].

Moreover, wood from tree pruning offers a wide spectrum of possibilities for their reuse and valorization, since they are still enriched in high added-value molecules, as foreseen by the circular economy [15]. Plants, in particular, produce secondary metabolites, which are not directly involved in the basic functions of growth, development, and reproduction of the organism but are essential for long-term survival and play multiple roles, including defense against predators or the attraction of pollinators. The above metabolites are endowed with numerous biological activities, making them also extremely important for human health and well-being. Moreover, due to their chemical and biological properties, applications for these metabolites have also been found in many other fields, serving as pigments, cosmetics, antioxidants, antifeedants, and so on [16].

Wood from plantation removal is usually produced when vines and olive or fruit trees are cleared out at the end of the lifetime of a plantation. In some cases, the termination of a plantation is driven by changes in the food market (in order to grow a new fruit or grape variety), by agricultural policies (for modernization and reconversion of plantations), or by other particular reasons (plague/disease, farmer, or exploitation manager). Considering the ha of different fruit trees and their half-life time, between 5–20 T dw/ha are discarded from trees, which may reach more than 2 MT yearly. As for pruning, the wood from plantation removals is mostly under-utilized in Europe [17], although the traditional use of firewood from the aerial part of the tree may be usual in some areas. In such cases, the stumps, roots, and thin branches are not used. In many cases, the whole tree is just uprooted, piled with others, and burned in the open air.

In summary, wood is a valuable waste product that is available in huge amounts every year, and its valorization deserves to be studied.

## 3. The Main Fruit Trees in Europe: Bioactive Compounds from Wood and Applications

Orchards could also be ‘exploited’ as a starting material through extracting secondary metabolites with high intrinsic values, due to their wide range of biological activities. For instance, several studies have demonstrated the potential uses of the secondary metabolites derived from agricultural wood extracts, including the antioxidant compounds present in vineyard wood and the radical scavenging properties of extracts derived from olive tree wood, as detailed below. In general, the following schema of wood compounds and applications can be established (Figure 3).

### 3.1. Olive Trees

Olive trees (*Olea europaea*) are small trees between 8 and 15 m tall with life expectancies of up to 1000 years, the oldest one dating from 2000 years ago [18]. Throughout history, the fruit of these trees has been appreciated and consumed as table olives and incorporated into bread, soups, salads, etc. In addition, oil obtained from these olives is one of the most consumed agricultural products around the world. In fact, the total harvested production of olives just for oil in the European Union in 2019 was 9.8 million tons, of which 5.6 million tons were collected in Spain [19]. This positions olive trees as one of the trees with the largest productive areas in the European Community, with Spain being the country with the highest productive area in the EU, as described in the introduction section. 

The production of olive wood discarded during olive pruning could vary from 3.5 to 10.5 kg/tree each year, which results in between 1.31 and 3.02 tons of discarded wood/ha [12,20]. In Spain, this is around 7 million tons of wood per year that is not leveraged. For the above reasons, many authors have investigated olive wood composition to propose extra uses for it.

#### 3.1.1. Bioactive Compounds from Olive Wood Waste

Olive trees contain a significant amount of carbohydrates, such as cellulose, hemicellulose, and lignin [21]. In fact, the olive tree has 32% more cellulose than other materials currently used in biorefinery, such as birchwood (26%), peanut shells (25%), or waste tea (17%) [22,23,24]. Within the carbohydrates group, different sugars have been identified in olive wood, in particular fructose, galactose, and glucose [25,26,27], explaining why it has been employed as an energy source in a multitude of situations. 

In addition, many interesting compounds have been isolated from olive tree wood, including lignans, fatty acids, nitrogen compounds, and iridoid [15,28,29,30]. These compounds are of great interest, as they exhibit properties related to plant and even human health. However, the quantification of these compounds in olive tree wood and, consequently, the evaluation of these properties has not been performed yet. 

Ortega-García et al. [26] extracted different phenolic compounds from *Olea europaea* L. cv. Picual wood using solid–liquid extraction. The extracts were analyzed by HPLC, and the results showed the presence of hydroxytyrosol (1500–2120 mg/kg dw), tyrosol (1720–3640 mg/kg dw), and oleuropein (22,300–58,020 mg/kg dw). These compounds have a crucial role in the nutritional and organoleptic properties of olives and olive oil. In addition, they present an important action in defending plants against pathogens and herbivores [31,32]. Finally, these three compounds have exhibited significant antioxidant properties, even preventing oil oxidation during storage and inhibiting the growth of Gram-positive microorganisms [33]. Pérez-Bonilla et al. [34] also evaluated the presence of different phenolic compounds from olive tree wood and analyzed their DPPH values (%). A total of six phenolic compounds were found: hydroxytyrosol (76.7%), tyrosol (3.7%), cycloolivil (19.7%), 7-deoxyloganic acid (6.8%), oleuropein (20.4%), and ligustroside (7.4%). Luján et al. [35,36] evaluated the presence of other phenolic compounds in wood from the Arbequina, Empletre, and Picual olive tree varieties using ultrasound- and microwave-assisted extraction coupled to analytical techniques, such as gas or liquid chromatography and mass spectrometry. The authors reported the following concentrations of phenolic compounds: verbascoside (1338–1681 mg/kg dw), apigenin-7-glucoside (10–16 mg/kg dw), and luteolin-7-glucoside (167–202 mg/kg dw). Faraone et al. [20] studied the presence of phenolic compounds in olive tree wood using GC-MS. The results exhibited a total phenolic compound content of 156.040 mg/kg dw and antioxidant activity of 188.840 mg Trolox equivalent (TE)/kg dw. In addition, 22 organic compounds (alcohols, aldehydes, and ketones), 13 phenolic acids, 2 benzenoid aromatics, 1 pyranone, and 3 amines were identified in olive trees. Allopurinol was the most concentrated organic compound. Lastly, Salido et al. [37] also analyzed the composition of wood extracts from 10 Spanish olive cultivars and found the following 16 phenolic compounds: (-)-olivil, (-)-olivil 4-*O*-β-d-glucopyranoside, 2″-hydroxyoleuropein, isojaspolyoside A, jaspolyanoside, jaspolyanoside isomer, jaspolyoside, jaspolyoside isomer, ligustroside, oleoside 11-methyl ester, (+)-1-hydroxypinoresinol 1-*O*-β-d-glucopyranoside, 7-deoxyloganic acid, eriodictyol, hydroxytyrosol, and tyrosol. The most abundant compounds were (-)-olivil and ligustroside (Table 1). However, as far as we know, these properties have not been evaluated in extracts obtained from olive tree wood.

#### 3.1.2. Applications from Olive Wood Waste

Olive wood is versatile and very tough, which is why it has been used for high-end indoor furniture, turned objects, veneer, carving, flooring, and high-end knife or tool handles, among others [18]. The first and main application for olive tree wood is biomass as a source of energy. Olive tree wood has been burned for years as a source of cooling or heating and electric energy, with important advantages in economic terms and with a considerable reduction in CO_2_ emissions [38,39]. In fact, many authors have based their research on the evaluation of the best pyrolysis conditions to optimize the energy and products obtained [40]. 

There are scarce alternative applications currently employed based on its bioactive compound content. Ateş et al. [41] evaluated the use of olive wood extract as an antifungal in vitro to propose it as a natural antifungal agent for different plants. To this end, the olive wood was powdered and extracted using the Soxhlet technique with methanol as the solvent. Four dosages of the extract, diluted with methanol, were added to a Petri dish that contained the fungus *Pleurotus ostreatus*. After eight days of incubation, the antifungal property (IC_50_) was measured. It was observed that the *P. ostreatus* growth was inhibited by 80% when the highest dose was used. The authors related these results with the presence of oleuropein, hydroxytyrosol, tyrosol, benzoic acid, *p*-coumaric acid, and catechin in the extract.

Tarfaya et al. [42] also evaluated the antimicrobial and antifungal properties in vitro of olive tree wood against different fungus and bacteria present in plants. The authors distilled the olive wood for 3 h before emulsifying the extract with an agar solution. Different doses of this mixture were confronted with six bacteria (*Escherichia coli*, *Staphylococcus aureus*, *Pseudomonas aeruginosa*, *Listeria monocytogenes*, *Enterococcus faecalis*, and *Klebsiella pneumoniae*) and three fungi (*Aspergillus niger*, *Aspergillus flavus*, and *Penicillium purpurogenum*). The results exhibited the capacity of the extracts to inhibit the concentration of all the studied bacteria and fungi. The best results were obtained for the inhibition of *K. pneumoniae* and *S. aureus*. Finally, the antioxidant capacity of the mixture was tested using a DPPH methodology. The results showed an antioxidant value of 1.45 ± 0.16 mg TE/mL, which is evidence of its interesting properties to be applied to the antifungal and antimicrobial treatment of plants.

Özkan et al. [43] used the Soxhlet methodology to obtain an extract from olive wood with methanol as the solvent. First, the antioxidant activity of the extracts was evaluated using CUPRAC, FRAP, and DPPH methodologies, obtaining 192.91, 105.23, and 116.05 mg TE/g dw, respectively, confirming the high antioxidant capacity of these extracts. After that, the extracts were tested against four Gram-positive bacteria (*L. monocytogenes* ATCC 7644, *Enterococcus faecium*, *S. aureus* ATCC 25923, and *Enterococcus durans*), four Gram-negative varieties (*Enterobacter aerogenes* ATCC 13048, *Salmonella kentucky*, *Salmonella typhimurium* SL 1344, and *E. coli*), and one fungus (*Candida albicans* ATCC 26555). All the extracts showed high antibacterial and antifungal activities, achieving the highest antibacterial activity against *L. monocytogenes*, which is a significant advance, as *L. monocytogenes* is one of the most virulent food-borne pathogens. Later, DNA protection against the hydroxyl radical damage generated by Fenton Reagent from the extract was also evaluated, exhibiting the capacity of the olive wood extract to inhibit the DNA damage arising for this reagent. Finally, olive wood extracts displayed significantly strong enzyme inhibition activity against acetylcholinesterase and butyrylcholinesterase enzymes, which are highly related to Alzheimer’s pathology and could be an important advance in its treatment. The authors concluded that these properties could be strongly related to the phenolic compounds present in olive wood, such as oleuropein, hydroxytyrosol, tyrosol, vanillin, protocatechic acid, benzoic acid, and *p*-coumaric acid, therefore suggesting that olive wood could be considered for use as a source of natural bioactive agents for dietary, pharmacological, and medical applications.

Finally, Zbidi et al. [44] discovered that oleuropein and (+)-cycloolivil in an olive tree wood extract reduced thrombin-induced platelet aggregation in vitro. The authors cut olive wood into chips and extracted the compounds by solid–liquid extraction with solvents of different polarities. The extract was analyzed and found to present a significantly high concentration of oleuropein and cycloolivil, which are known to present an important inhibitory effect of platelet aggregation. For this reason, the authors evaluated the effect of olive wood extracts on platelets obtained from healthy and diabetic volunteers. The inhibitory effect of the extract was slightly greater in the platelets from the healthy individuals than in those from the diabetic patients, but when concentrations of 100 μM and over were used, thrombin-evoked platelet aggregation was similar in both groups. This is a significant advance because this by-product could be employed to prevent thrombotic complications associated with platelet hyperaggregability and used in antiaggregant therapeutic strategies in illnesses, such as type 2 diabetes. A summary of the bioactive compounds isolated from olive wood and their applications can be found in Table 1.

### 3.2. Grapevines

According to the International Organization of Vine and Wine (OIV), the consumption of wine in the world during 2016 was 244,328,000 hL, 146,790,000 hL of that in European countries [45]. This means that a total of 3.2 million ha were devoted to vines in Europe in 2015, of which only 60,351 ha were used to cultivate table grapes, the main purpose being wine production (more than 78%) [46].

During grapevine pruning, usually for wine production, grapevine canes (also called vine shoots) are the major by-products, with around 2–5 tons/ha each year [47], which is a huge amount of discarded wood.

#### 3.2.1. Bioactive Compounds from Grapevine Cane Waste

Grapevine canes have been shown to have a high content of carbohydrates, with a total concentration of 3496–44,200 mg/kg dw [48], which is why they have been suggested for use as hybrid particleboards. In addition, canes have exhibited significant concentrations of some minerals that are of interest to human health, such as K (5190–7020 mg/kg dw), P (420–930 mg/kg dw), Ca (5950–10,210 mg/kg dw), Fe (2.6–6.8 mg/kg dw), Mg (19.4–111.2 mg/kg dw), and Zn (7.0–98.4 mg/kg dw) [48].

However, the compounds in grapevine canes that have been studied in the greatest depth are the phenolic compounds, in particular stilbenes. Ferreyra et al. [49] analyzed the phenolic compounds in canes of the Malbec variety. The authors identified and quantified the main phenolic compound families: phenolic acids (gallic acid, syringic acid, cinnamic acid, caftaric acid, and *p*-coumaric acid), stilbenes (*trans*-resveratrol and ε-viniferin), flavanols (procyanidin B1, catechin, procyanidin B2, (−)-epicatechin, (−)-gallocatechin gallate, (−)-gallocatechin, and (−)-epigallocatechin gallate), flavanones (naringin and naringenin), flavanonol (astilbin), and flavonols (quercetin-3-glucoside, quercetin-3-galactoside, kaempferol, and myricetin). Those with the highest concentrations were: (+)-catechin (101–8320 mg/kg dw), quercetin-3-glucoside (191–2651 mg/kg dw), and caftaric acid (424–5488 mg/kg dw).

Techniques, such as Soxhlet extraction or ultrasound-assisted extraction have been commonly used for these analyses. Barros et al. [50] analyzed seven Portuguese canes of different varieties, detecting flavanols (epicatechin, epicatechin gallate, procyanidin dimer A, procyanidin dimer B epigallocatechin-(epi)catechin dimer, and procyanidin dimer gallate), flavonols (isorhamnetin-3-*O*-(6-*O*-feruloyl)-glucoside, quercetin-3-*O*-rutinoside, quercetin-3-*O*-glucoside, kaempferol-3-*O*-rutinoside, and kaempferol-3-*O*-glucoside), hydroxycinnamic acids (caftaric, coumaric, coutaric, and ferulic acids), anthocyanins (malvidin-3-*O*-glucoside, malvidin-3-*O*-(6-*O*-caffeoyl)-glucoside, and malvidin-3-rutinoside), and stilbenes (ε-viniferin).

Other authors [51] summarized the phenolic concentrations in grapevine canes with the different analytical techniques employed. Six families were evaluated (hydroxybenzoic acid, hydroxycinnamic acid, flavanols, flavones, flavononols, and tannins). In general, the highest concentrations were found for flavonols, in particular quercetin-3-*O*-glucoside (200–126,800 mg/kg dw) and quercetin-3-*O*-rutinoside (900–41,800 mg/kg dw).

The family of phenolic compounds that has attracted the attention of many authors is stilbenes, due to their interesting properties related to both plant and human health. *Trans*-resveratrol and *trans*-ε-viniferin were the first stilbenes to be isolated from grapevine canes in 1999. Since then, many authors have evaluated their presence in numerous grapevine cane varieties, with *trans*-resveratrol being the stilbene with the highest concentration in most grapevines, ranging between 11–66,200 mg/kg dw, followed by *trans*-ε-viniferin (47–40,600 mg/kg dw) and r2-viniferin (60–15,200 mg/kg dw) [52,53,54].

#### 3.2.2. Applications of Grapevine Cane Waste

Grapevine canes have been used as an energy source or in the production of biorefinery compounds or bioactive carbon [55,56]. Due to their high lignin content, their compaction with less compressive strain, and high density they have also been employed for hybrid particleboards [57]. However, the main application for grapevine canes is as compost on the field, because they are rich in lignin, cellulose, nitrogen, and potassium, which are compounds of great interest for the vineyard.

Based on the antimicrobial activity of many of the compounds detected in the wood, grapevine cane extracts were evaluated as antifungal compounds, and a significant reduction in the downy mildew (*Plasmopara viticola*) attack and infection in greenhouses and vineyards was found [58,59]. Furthermore, no adverse effects were observed on the grapevine or auxiliary fauna. Other authors [60] studied the use of different grapevine extracts (from canes, roots, and trunks) as natural plant agents to combat *P. viticola*, one of the major fungal pathogens in grapevines. The results showed that trunk extract, followed by root extract, showed the highest antifungal activities. Moreover, these extracts also showed a significant insecticidal effect, inhibiting the larval development and food intake of *Leptinotarsa decemlineata* (Colorado potato beetle) and *Spodoptera littoralis* (African cotton leafworm) [61,62]. Consequently, grapevine wood extract could be proposed as a natural insecticide and fungicide for the vineyard itself or even for different crops.

Moreover, grapevine cane extracts have been tested as biostimulants due to their high content of phenolic compounds, carbohydrates, and minerals. Aqueous extracts from grapevine canes were applied in grapevines. The results showed an increase in the amount of gallic acid, hydroxycinnamoyl tartaric acid, acylated anthocyanins, flavonols, and stilbenes in the grapes [63,64]. In addition, Cebrián-Tarancón et al. [65,66] tested the use of toasted cane fragments (chips or granules) in winemaking. No significant differences in the main enological parameters were found in any case, but significant differences in both the phenolic acid and stilbene concentrations were observed, which may have important implications for the health of consumers.

Finally, based on their high antioxidant and antimicrobial properties, grapevine cane extracts have been proposed as an alternative to SO_2_ in winemaking to mitigate the health-related problems associated with SO_2_ use [67]. Grapevine cane extracts with different stilbene concentrations have been evaluated. Extracts with a low stilbene percentage (29%) proved their antioxidant capacity in comparison with other alternatives currently used in the market; however, modifications to the organoleptic properties were perceived by consumers [68,69,70]. Extracts with a higher percentage (45%) still showed some side effects on wines. Lastly, an extract with a 99% concentration of stilbenes (ST99) was evaluated in a wine matrix, showing promising properties against spoiled microorganisms related to wine [71]. Moreover, a toxicological study on ST99 proposed this extract as a promising extract, due to its antioxidant properties, presenting no genotoxic effects under in vivo studies [72].

In summary, grapevine canes are a source of interesting compounds that can be employed in applications ranging from energy production to the agri-food industry and even the treatment of several plant infections. A summary of the bioactive compounds in grapevine canes and their concentration ranges can be found in Table 2.

### 3.3. Nut Trees

Focusing on nuts, almonds (*Prunus dulcis*) and walnuts (*Juglans regia*) represent over 35% (> 318,000 tons) and 34% (> 305,000 tons) of the European production, respectively (Figure 4). Hazelnuts (*Corylus avellana*) and European chestnuts (*Castanea sativa*) are also produced in large amounts, representing 17% (> 155,000 tons) and 13% (> 106,000 tons) of the production, respectively. To a lesser extent, pistachio nuts (*Pistacia vera*) are also cultivated in Europe (1%, 12,000 tons) [73,74,75,76,77].

The annual management of mature nut trees, which are spread over more than 1,000,000 hectares in Europe, generates a huge amount of wood residues through lignified suckers, dead wood, and diseased or badly oriented tree shoots [78]. Regular pruning work gives rise to pruned wood, while the more sporadic old-orchard tree removal practices engender the availability of wood trunks. According to data from Huang and Lapsley [79] on almond orchards, it could be estimated that the pruning biomass is about 2.4 tons/ha per year. Another study calculated an amount of pruned wood from almond trees of 1.6 tons/ha per year, which agrees with the previously reported data [80]. In addition, taking into account that between 9000–20,000 ha of almond orchards are removed each year, a total of wood waste of 63.4 tons/ha was obtained [79]. In hazelnut orchards, between 1.4 and 2.7 tons/ha of pruned wood were estimated [78,81]. Regarding chestnut groves, a study showed that pruning practices produced between 22 and 33 tons/ha of wood residues, while a more recent study highlighted a pruned wood amount between 3.9 and 24.9 tons/ha, these differences are due to variations in tree age, site density, and pruning intensity [82,83]. Examining walnut orchards, Bilandzija et al. [80] reported that pruning management generated 0.5 tons/ha, which was inferior to the other nut trees. Overall, the mean amount of pruned wood from nut trees is close to 2 tons/ha per year, which could vary substantially in chestnut (highest value) and walnut (lowest value) orchards.

The following sections deal with the chemical compounds as well as the bioactivities involved in the agri-food domain of almond, walnut, and chestnut wood biomasses. Unfortunately, due to the absence of literature data, the valorization of hazelnut and pistachio pruned wood is not discussed in the current review.

#### 3.3.1. Almond Wood Waste

##### Bioactive Compounds from Almond Wood Waste

Pruned residues from almond trees are predominantly composed of polyphenols, mainly phenolic acids, such as 4-hydroxybenzoic acid, protocatechuic acid, salicylic acid, vanillic acid, and syringic acid. To a lesser extent, the almond wood biomass also contains ferulic acid and *p*-coumaric acid, as well as the cinnamaldehydes sinapinaldehyde and coniferyl aldehyde.

Focusing on polysaccharide content, almond wood waste contains 22.3% glucan (glucose polymer) and 11.1% xylan (xylose polymer), making a total of 33.4% (dried biomass) fermentable sugars. Furthermore, almond pruning residues were also composed of lignin with a value of 20.0% of biomass [84].

##### Application of Almond Wood Waste in the Agri-Food Industry 

Few data on the bioactivities of almond pruned wood were available in the literature, but some studies investigated the properties of phenolic acids present in woody biomass and isolated from other almond by-products, such as the seeds or aerial parts (stems and leaves). In this sense, the 4-hydroxybenzoic, protocatechuic, and vanillic acids isolated from almond seeds showed an antibacterial effect against two Gram-positive bacteria (*S. aureus* and *Bacillus subtilis*) and five Gram-negative bacteria (*P. aeruginosa*, *S. typhimurium*, *Proteus vulgaris*, *E. coli*, and *K. pneumoniae*) [85]. The protocatechuic acid isolated from almond skin also inhibited *Helicobacter pylori* (MIC range, 128 to 512 mg/L), which is a Gram-negative bacterium known to be a gastric pathogen of humans (responsible for chronic gastritis, peptic ulcers, intestinal metaplasia, and cancer) [86]. Moreover, protocatechuic acid and *p*-hydroxybenzoic acid, followed by vanillic acid and syringic acids, exhibited antioxidant capacities through a DPPH assay with values from 4.5 to 12.9 µmol TE/g [87]. Vanillic acid was also involved in the inhibition of the β-glucosidase enzyme, which is involved in various metabolic processes and is part of promising therapeutics in the treatment of related diseases [88]. Overall, almond wood waste, rich in polyphenols, especially phenolic acids, might be valorized for medical applications, in cosmetic products, or as food additives through their antibacterial and antioxidant properties.

#### 3.3.2. Walnut Wood Waste

##### Bioactive Compounds from Walnut Wood Waste

In wood from pruned walnut trees, polyphenols were the main compounds, especially the phenolic acids gallic acid, ellagic acid, and ferulic acid, as well as the flavonol quercetin-3-β-d-glucoside [89]. In a water extract of pruned wood, gallic acid reached 98.0 mg/kg, ferulic acid reached 48.2 mg/kg, and quercetin-3-β-d-glucoside reached 38.3 mg/kg. In an ethanolic extract, the major compounds were quercetin-3-β-d-glucoside, with 150.2 mg/kg, and ellagic acid, with 101.4 mg/kg [89]. In a methanolic extract of walnut sawdust, the amount of ellagic acid reached 393.0 mg/kg (Table 3) [90].

Wood from pruned walnut trees also contained other phenolic acids, such as chlorogenic, neochlorogenic, *p*-coumaric, protocatechuic, caffeic, vanillic, syringic, and salicylic acids [84,89,91]. Other flavonoids were also characterized, such as kaempferol, quercetin, epicatechin, and procyanidin B1 and B2 [89,91]. In addition, this wood was composed of phenolic aldehydes (vanillin, 4-hydroxybenzaldehyde and 3,4-dihydroxybenzaldehyde) and cinnamaldehydes (coniferyl aldehyde, sinapinaldehyde, and 4-hydroxycinnamylaldehyde) (Table 3) [84,91].

Apart from polyphenols, walnut pruned wood contained polysaccharides, with 26.5% glucan, 10.6% xylan, and 20.0% lignin content [84]. Several organic compounds were also identified by GC-MS, such as elemicin, which is a phenylpropene, the terpenoid curzerene, *trans*-benzylidenacetone, and the esters butyl palmitate and isobutyl stearate (Table 3) [91].

##### Application of Walnut Wood Waste in the Agri-Food Industry 

Examining the antioxidant activities of pruned wood, Fernández-Agulló et al. [89] tested aqueous and hydro-ethanolic extracts through DPPH, FRAP, and ABTS assays. The extracts demonstrated a radical scavenging activity (EC_50_ values of 95–180 mg/L), reducing power (0.9–1.7 µmol ascorbic acid eq/mg), and lipid peroxidation inhibition (EC_50_ values of 757–1232 mg/L).

Another study investigated the antifungal property of wood trunk of walnut against *Trametes versicolor*, a white-rot fungus involved in poplar wood disease. Samples of poplar wood were impregnated with walnut wood extract, followed by exposition to *T. versicolor* for 14 weeks. The lowest wood weight loss was observed in the samples treated with a 1.5% concentration of extractive solution, obtaining value of 30.4%, while the highest weight loss was measured in the untreated control (36.9%). Therefore, the authors concluded that walnut wood was able to slow down the effect of the fungus attack [92].

Furthermore, a recent study investigated the evolution of the phenolic composition and sensory characteristics of a red wine stored for 30 days in contact with toasted chips from walnut, a non-conventional wood species in enology. The wine kept in contact with walnut chips showed significantly higher values for the oligomeric fraction of proanthocyanidins, whereas, from a sensory point of view, a tendency for lower scores for most sensory descriptors was obtained [93].

Apart from studies developed on walnut extracts, the major compounds of walnut wood were also tested through their isolation from wood or other walnut by-products. Regarding immunomodulatory function, a study highlighted that gallic acid and ellagic acid decreased lymphocyte T-cell activation/growth and the secretion of interleukins. These data suggest that supplementation of gallic and ellagic acids to walnut-containing foods might ameliorate walnut-induced allergic reaction [94]. Related to inflammatory diseases, gallic acid exhibited inhibitory activity on the nitric oxide production of macrophages. Nitric oxide is one of the most important reactive nitrogen species and constitutes an important target in the inflammatory response/anti-inflammatory treatment [95]. Gomes et al. [96] argued also that ellagic acid glycoside and quercetin derivatives from walnut may exert an antibacterial effect against *S. aureus* and may be proposed as promising agents for the reduction of the occurrence of dairy food industry contaminations.

#### 3.3.3. Chestnut Wood Waste

##### Bioactive Compounds from Chestnut Wood Waste

Among all nut trees, chestnut wood is certainly the most studied, and wood trunk extracts are already commercially available (Farmatan^®^, Tanex^®^, and Silvafeed^®^). This success and this recognition are due to its composition of polyphenols and, more precisely, in hydrolysable tannins. Among hydrolysable tannins, two main classes are distinguished, ellagitannins, which contain a hexahydroxydiphenoyl unit and give rise to ellagic acid by hydrolysis, and gallic acid derivatives [97]. Owing to recent technological advances, especially in mass spectrometry, the deciphering of the chemical composition of chestnut wood is more developed. In recent studies, several ellagitannin derivatives were identified, such as ellagic acid, ellagic acid pentoside, vescalin, vescalagin, castalin, castalagin, punicalagin, pedunculagin, roburin A-E, (mono-, di- tri-, tetra-, and penta-)galloyl glucose, valoneic acid dilactone, and castavaloninic acid [97,98]. Gallic acid was also characterized. In addition to these compounds, other ellagitannins and gallic acid derivatives were putatively identified with the proposed structures supported by molecular formula information, HR-MS fragmentation, and UV data [97]. From a quantitative point of view, the phenolic fraction of chestnut wood extracts could vary between 2–4% of dw, depending on both the solvent and temperature employed for the extraction [99] (Table 3).

Apart from polyphenols, some terpenes were also identified in chestnut wood, such as the triterpenoids bartogenic acid, bartogenic acid 28-*O*-β-d-glucopyranosyl ester, chestnoside A, and chestnoside B [100]. Using subcritical water extraction, Alarcon et al. [101] reported that chestnut wood not only contained flavoring compounds, such as terpenoids and norisoprenoids (dehydrovomifoliol), but also lactones (β-methyl-γ-octalactones), aliphatic aldehydes (nonanal), and benzenic (vanillin derivatives) compounds.

##### Application of Chestnut Wood Waste in the Agri-Food Industry 

Over the past five years, the biological activity of chestnut wood remains an attractive and highly studied topic, owing to the presence of hydrolysable tannins and other compounds. Regarding the antioxidant abilities of chestnut wood extracts, Faraone et al. [102] showed a high reducing ability of radical DPPH (1234 mg TE/g) and ferric reducing antioxidant power (2263 mg TE/g). Additionally, Tomažin et al. [103] studied the oxidative stability of meat by adding chestnut wood extract to the diet of pigs. Among various indicators, the main effects observed were an increase in the water holding capacity and a positive color evolution of dry-cured bellies, due to the antioxidant supplementation. Similarly, Liu et al. [104] investigated the use of chestnut wood extract as an alternative feed additive in the broiler diet and demonstrated increases in antioxidant status, cholesterol metabolism, and growth performance without affecting normal meat quality. Molino et al. [105] also exhibited that chestnut wood positively modulated the gut microbiota using an in vitro digestion-fermentation assay.

Furthermore, another study showed the antifungal activity of chestnut wood extract against different filamentous fungi of agronomical and food interest, such as telluric phytopathogens (*Fusarium oxysporum*, *Fusarium solani*, *Rhizoctonia solani*, and *Sclerotium rolfsii*) and post-harvest pathogens (*Botrytis cinerea*, *Penicillium digitatum*, and *Penicillium italicum*) [106]. Hydrolysable tannins from chestnut wood extract also showed antibacterial activities by inhibiting the growth of *S. aureus*, *Streptococcus uberis*, *P. aeruginosa*, *Streptococcus agalactiae*, *E. coli*, and *K. pneumoniae* (concentration > 630 mg/mL) [107].

Chestnut wood barrels have also been used in the aging of wine brandies. In fact, a transfer of compounds occurs between the wood and beverage, influencing the physicochemical and sensory properties of the wine. Wine aged in chestnut barrels is distinguished by a higher phenolic content and higher levels of some low molecular weight compounds, such as gallic acid, vanillic acid, vanillin, and volatile phenols [108].

Apart from polyphenols, other compounds from chestnut have been also studied. In fact, the cytotoxicity of triterpenoids isolated from *C. sativa* wood was investigated against two cancer cell lines (PC3 and MCF-7). The results showed that the terpene chestnoside B exhibited a stronger cytotoxicity (IC_50_ of 12.3 mM) towards MCF-7 cells than positive controls, highlighting the potential of triterpenoids from chestnut wood to prevent breast cancer [100].

Even though chestnut wood is commercialized nowadays, especially in the field of enology, and may not be considered agricultural waste to such an extent, other applications should be promoted. Chestnut wood could be valorized for pharmaceutical purposes or as a food additive, due to its antioxidant, anti-carcinoma, antifungal, and antibacterial properties.

### 3.4. Apple and Pear Trees

According to European statistics, the pome fruits are the fourth most common species cultivated around the European Union, covering over 600,000 ha, with around 9 MT being produced every year. Among the pome genus, apples and pears are the main fruits consumed. Apples represent 80% of the total pome production, and pears the remaining 20% (Figure 5).

At pruning, between 2–5 tons/ha are produced from apple and pears trees each year. Up to 6.8 and 5.6 kg wood/ha have been described in the Czech Republic [109]. If we add the fallen wood from climatic events or fruit harvesting processes, there is a huge amount of wood produced as a by-product that is not given extra value.

#### 3.4.1. Bioactive Compounds from Apple and Pear Wood Waste

Some authors have evaluated the availability of nutrients in pome tree wood. Apple tree wood showed a higher macronutrient content, with a concentration of 6810 and 13,900 mg/kg of N and Ca available, respectively. Meanwhile, pear tree wood showed a higher micronutrient content, with 144 and 118 mg/kg of Fe and B, respectively [110]. The amount of carbohydrates has been also evaluated in apple wood, detecting the presence of starch, sorbitol, sucrose, and glucose or fructose, with concentrations from 3 up to 58 mg/g dw [111]. In addition, interesting values of organic and fatty acids, alcohols, lignin, and cellulose have been detected in apple tree wood [112]. However, to the best of the authors’ knowledge, these compounds have not been isolated or identified in wood from pear trees yet. Few researchers have focused their studies on the bioactive compounds in apple tree wood. Withouck et al. [113] analyzed the composition of wood from *Malus domestica* ‘King Jonagold’ apple trees. The results exhibited a total phenolic compound concentration of 9600 mg/kg dw and a total flavonoid content of 2570 mg/kg. After ultrasound-assisted extraction, the following families were identified: flavanols ((−)-epicatechin gallate, (−)-epicatechin, (+)-catechin, procyanidin B1 and B2), phenolic acids (gallic acid, caffeic acid, vanillic acid, *p*-coumaric acid, and ferulic acid), flavonols (rutin, quercetin, and kaempferol-3-glucoside], flavanones (naringin and naringenin), and dihydrochalcone (phloridzin and phloretin). The total phenolic compound concentration reached up to 46,600 mg/kg dw. In addition, the obtained extracts exhibited a significant antioxidant activity, with a DPPH assay of 28.4 mg TE/g dw, and a significant antimicrobial property, with a 68% reduction in the growth of *Enterococcus faecalis*.

Finally, Moreira et al. [114] used microwave-assisted extraction (MAE) for the analysis of *Malus domestica* apple tree wood. The authors reported a total phenolic and total flavonoid concentration of 2670 mg/kg dw and 1000 mg/kg dw, respectively. The authors analyzed the extracts obtained by HPLC-PDA and identified a total of 18 phenolic compounds in the apple wood: (+)-catechin, kaempferol, kaempferol-3-*O*-glucoside, kaempferol-3-*O*-rutinoside, myricetin, naringenin, naringin, phloretin, phlorizin, quercetin, rutin, 4-hydroxyphenilacetic acid, chlorogenic acid, cinnamic acid, ferulic acid, gallic acid, protocatechuic acid, sinapic acid, and resveratrol. Phlorizin was the most abundant phenolic compound, with concentrations ranging from 6890 to 8770 mg/kg dw.

#### 3.4.2. Applications of Pome and Pear Wood Waste in the Agri-Food Industry 

Apple and pear wood waste has been employed principally as firewood to obtain heating or electric energy [115] and as biochar [116], since it presents the highest net calorific value (around 15.7 MJ/kg) in comparison with other woods [104,109]. Moreover, this wood has also been burned to produce biocarbon or vinegar [117,118,119]. The direct application of these woods in the field is its degradation to contribute certain nutrients to the soil [120].

Moreover, apple tree wood has been used in the production of particleboards, with structural applications, and for furniture production [121]. Another study relied on the lignocellulosic content of apple and pear wood waste, among others, to demonstrate its worth as a substrate for growing the oyster mushroom, *P. ostreatus*, one of the most edible mushrooms because of its nutritional quality, ease of cultivation, and economic potential. *P. ostreatus* is a saprotroph species with the ability to degrade cellulose, playing an important ecological role in the disintegration of organic waste. The authors concluded that pruning residues from apple trees and pear trees can be successfully used as raw materials to obtain substrates for *P. ostreatus* cultivation [120]. Pear wood waste, containing a considerable amount of cellulose, may be useful to produce paper, plastics, photographic film, magnetic tapes, protective coatings, and electrical parts [122].

In addition, antioxidant activity has been attributed to apple wood extracts. The antioxidant activity of acetone/water extracts was evaluated by the DPPH method, reaching values of 80% (% reduction/g dry weight), which significantly proves its high antioxidant activity [113]. The same authors also researched the antimicrobial effect of apple wood extracts. Bark extracts obtained by acetone/water mixture presented the highest phenolic (22.84 mg GAE/g DW) and antioxidant activity (1.07 mM FeSO_4_.7H_2_O/g dw for the FRAP assay), while the extract also inhibited the growth of the Gram-positive bacteria *E. faecalis* and *S. aureus* by 100% [113]. Moreover, phlorizin, the most abundant phenolic compound in wood extracts, is able to lower glucose plasma concentrations independent of insulin [123]. Therefore, the authors suggest that these antioxidant extracts can be used in the food industry, pharmaceutics, and cosmetics. A summary of the bioactive compounds isolated from apple wood and their applications can be found in Table 4.

### 3.5. Stone Fruit Trees

In Europe, the stone fruits are the fifth most cultivated common fruit in terms of surface area (0.6 million hectares of land) and the fourth most produced (7.3 MT). Examining the proportions of each stone fruit, peaches (*Prunus persica* L.) are the most produced, accounting for 42%, followed by plums (*Prunus domestica* L.) and nectarines (*Prunus persica* var. *nucipersica*), which account for 20% and 16% of the European production, respectively. Finally, cherries (*Prunus avium*) and apricots (*Prunus armeniaca*) also represent an important part of stone fruit production, accounting for 13% and 9%, respectively (Figure 6) [124].

Similar to other common fruit species, stone fruit production is accompanied by the generation of wood residues, mainly by annual pruning and, to a lesser extent, by the renewal of the orchard. Several studies of peach groves mentioned consensual values of 2.5, 2.7, and 2.9 tons of pruned wood/ha per year [80,125,126]. Similar values were reported for nectarine orchards [80]. On the contrary, heterogenous data on plum trees are reported by Pari et al. [126], who estimated 0.8 tons/ha per year of pruned wood, by Bilandzija et al. [80], who estimated 2.1 tons/ha per year, and by Burg et al. [109], who estimated 4.6 tons/ha per year [80,109,126]. In cherry orchards, previous reports showed that annual pruning results in between 1.8 and 2 tons/ha per year of wood residues, while in apricot groves the amounts range between 1.2 and 1.9 tons/ha per year. Globally, stone fruit trees appear to produce around 2 tons/ha per year of wood waste.

The following sections deal with the chemical compounds and bioactivities relevant to the agri-food domain of peach, plum, cherry, and apricot woody biomasses. Unfortunately, due to the absence of literature data, nectarine wood valorization is not discussed.

#### 3.5.1. Peach Wood Waste

##### Bioactive Compounds from Peach Wood Waste

The chemical composition of pruned wood from peach orchards was studied by Nakagawa et al. [127], who attempted to isolate the active compounds from an ethanol extract. The following polyphenols were identified: two flavanols, catechin and 4′-*O*-methylcatechin; the flavonol quercetin 3-*O*-β-d-glucopyranoside; the chalcone designated as 4,2′,4′-trihydroxy-6′-methoxychalcone 4′-*O*-β-d-glucopyranoside; the phenolic acid ferulic acid; and a phenolic compound called phenyl *O*-β-d-glucopyranoyl-(1→6)-β-d-glucopyranoside. In addition, two pentacyclic triterpenoids named oleanolic acid and ursolic acid were also characterized.

Apart from secondary metabolites, peach wood was composed of 42.4% crude fiber, such as cellulose or lignin, 3.9% crude protein, and 1.8% ethereal extract, which contains fats and fatty acids [120].

##### Application of Peach Wood Waste in the Agri-Food Industry

From the biological investigation, wood waste from peach tree pruning and its major compounds were studied with a view towards understanding their antioxidant, anti-lipase, and anti-dementia activities. The free radical scavenging capacities of water and ethanolic extracts were tested using the ORAC assay and showed values of 0.2 and 3.0 mg TE/mg, respectively. The flavonol quercetin 3-*O*-β-d-glucopyranoside was shown to be the main antioxidant compound. The anti-lipase activity, which is linked to the fight against obesity, exhibited IC_50_ values of 54.9 µg/mL, with oleanolic acid as the main contributor. The neuroprotective activity of ethanolic extract was evaluated through its acetylcholinesterase inhibition capacity, and IC_50_ values of 83.4 µg/mL were found. In this case, 4′-*O*-methylcatechin, chalcone derivative and ferulic acid were stated as the main active compounds [127].

Aiming to discover a new natural antihypertensive agent, Kim et al. [128] tested the vasorelaxant effect of pruned wood extract from peach trees and studied its associated mechanisms. Their findings demonstrated that ethanolic extract caused a concentration-dependent vasorelaxation of rat aortic rings. Peach wood waste could thus be valorized as a functional food for hypertension treatment and could be implemented in nutraceutical or pharmaceutical areas [128].

Another study into the lignocellulosic content of peach tree shoots demonstrated that peach wood could also be used in agriculture to promote the development of *P. ostreatus* [120].

#### 3.5.2. Plum Wood Waste

##### Bioactive Compounds from Plum Wood Waste

A chemical study of pruning residues from plum trees was performed recently, highlighting the presence of polyphenols. Among them, two flavanols (catechin and epicatechin), one phenolic glucoside (annphenone), and six dimeric A-type proanthocyanidins ((−)-ent-epicatechin-(2α→O→7,4α→8)-catechin, (−)-ent-epicatechin-(2α→O→7,4α→8)-epicatechin, (−)-ent-epiafzelechin-(2α→O→7,4α→8)-catechin, (+)-epiafzelechin-(2β→O→7,4β→8)-epicatechin, (+)-epiafzelechin-(2β→O→7,4β→8)-afzelechin, and (−)-ent-epiafzelechin-(2α→O→7,4α→8)-epiafzelechin) were unambiguously characterized. Quantitatively, the proanthocyanidins were the main compounds, with a total content of 8.5 g/kg dw, followed by annphenone with 2.3 g/kg, and the flavan-3-ols with 1.4 g/kg [129]. The pruned wood also contained fibers with respective lignin and cellulose contents of 33.6% and 53.2% [130].

##### Application of Plum Wood Waste in the Agri-Food Industry

Biologically, pruned plum tree wood showed antimicrobial and antibiofilm activities against several foodborne microorganisms (*Salmonella enterica*, *E. coli*, *S. aureus*, *L. monocytogenes*, *Bacillus cereus*, *Enterococcus casseliflavus*, *E. faecium*, *Staphylococcus saprophyticus*, *Lactobacillus casei*, *Pantoea agglomerans*, *Klebsiella terrigena*, *Enterobacter* sp., and *Salmonella* sp.). Among them, the most significant antimicrobial effect was evaluated on the *Enterobacter* sp. strain, which is characterized by high tolerance to different biocides. All the isolated compounds described above showed MIC values of 100 mg/L against this resistant bacterium. These results point to the potential paths for the development of a food preservation plan using plum wood biomass as a biocide for the food industry or even as a food preservative [129].

It could be noted that the extract from plum wood trunk was also studied in brandy aging through analyzing the physical-chemical and organoleptic indices of this beverage and its consumer properties. In association with oak wood, the use of plum wood increased the quality and the organoleptic evaluation of the beverage, suggesting it could be used in the enology and beverage industry [131].

#### 3.5.3. Cherry Wood Waste

##### Bioactive Compounds from Cherry Wood Waste

Chemically, the pruned wood from cherry trees is mainly composed of polyphenols. The phenolic compounds belong mainly to the family of flavonoids, more specifically, to the sub-classes of flavanols (catechin), flavanonols (taxifolin, aromadendrin, and aromadendrin-7-*O*-glucoside), flavanones (naringenin, naringenin-7-*O*-glucoside, sakuranetin, sakuranetin-5-*O*-glucoside, pinocembrin, and liquiritin), and flavones (chrysin, tectochrysin, dihydrowogonin, and apigenin) [132]. The phenolic acid benzoic acid and the anthocyanin pelargonidin-3-*O*-glucoside have also been identified [133]. Apart from polyphenols, other metabolites were characterized, including carboxylic acids, fatty acyls, organooxygen compounds, and imidazopyrimidines. Some amino acids, such as asparagine, proline and glutamine, as well as the polyol mannitol, were notably abundant. Overall, more than 400 compounds were listed in pruned wood from cherry orchards [133].

The trunk wood of cherry trees has also been well-characterized, demonstrating a similar composition, with the presence of various bioactive phenolics, such as phenolic acids, flavonols, flavones, isoflavones, flavanones, flavanonols, stilbenoids, and coumarins [134,135,136]. The main compounds were the flavanonol taxifolin (8455.7 mg/kg), followed by the flavanones pinocembrine (1851.1 mg/kg) and naringenin (409.2 mg/kg), the flavone chrysin (716.1 mg/kg), and the phenolic acid ellagic acid (267.7 mg/kg) [134]. The cherry wood was also characterized by the presence of isoflavones, such as genistein and daidzein [137]. Volatile compounds were also investigated in a subcritical water extract. Terpenoid and norisoprenoid compounds (3-oxo-α-ionol, 3-oxoretro-α-ionol, and dehydrovomifoliol), alcohols and aliphatic aldehydes (nonanal and 3-octen-1-ol), aromatic compounds (vanillin, isoeugenol, sinapinaldehyde, and coniferaldehyde), and traces of lactones were identified and quantified (Table 5) [101].

##### Application of Cherry Wood Waste in the Agri-Food Industry 

Pruned cherry waste and its major compounds were recently tested for their antimicrobial and antibiofilm activity against strains from type culture collections (*S. enterica*, *E. coli*, *S. aureus*, and *L. monocytogenes*), as well as on multi-resistant strains (*B. cereus*, *E. casseliflavus*, *E. faecium*, *S. aureus*, *S. saprophyticus*, *L. casei*, *P. agglomerans*, *K. terrigena*, *Enterobacter* sp., and *Salmonella* sp.). The flavanol catechin, the flavanonols taxifolin and aromadendrin, the flavanone pinocembrin, and the flavone tectochrysin showed the highest antimicrobial activities, with MIC values of 100 µg/mL. In addition, taxifolin was able to inhibit the formation of biofilm (at 1 µg/mL) by *Enterobacter* sp. and caused the disruption of preformed biofilm (at 0.1 µg/mL) [132]. Due to its inhibitory activity against various foodborne microorganisms, pruned cherry wood might, thus, be of value as an antimicrobial agent or food preservative for the food industry.

Regarding wood from cherry trunks, its antioxidant capacity was evaluated using a DPPH assay and showed values of 299.9 mmol TE/kg [134]. Its antimicrobial activity was also demonstrated with a high level of growth inhibition against *S. aureus* and *Streptococcus mutans*, a moderate inhibition of *E. faecalis*, and a significant bactericidal ability towards *C. albicans* and *L. monocytogenes* [137].

Furthermore, trying to find new materials to endow their wines with a special personality, enologists investigated wood to substitute or complement the commonly used oak wood. In this sense, wood from cherry tree trunks was studied in wine and vinegar aging. Rich in flavonoids, compounds, such as eriodictyol, sakuranetin, pinocembrin, and chrysin, were transferred to the wines and may represent potential phenolic markers of the use of this wood species. Moreover, the use of cherry wood barrels promoted a faster evolution of catechin, procyanidins, and flavonols, with a condensation phenomenon that allowed for the stabilization of both the tannins and pigments of aged red wines [138]. During vinegar aging in cherry wood barriques, the flavanonol taxifolin was transferred to the vinegar and can be considered a marker of vinegar aged in cherry wood. From a sensory point of view, aging in cherry wood increased markers for red-fruit attributes [139]. Overall, the use of cherry wood in enology has resulted in greater diversity and wines with specific physical, chemical, and sensory variations.

#### 3.5.4. Apricot Wood Waste

##### Bioactive Compounds from Apricot Wood Waste

The chemical composition of apricot wood from pruning practices was studied by Bruno et al. [140]. They estimated a total polyphenol content between 153.3–188.0 mg of gallic acid equivalent (GAE)/g, depending on the extraction method. Using GC-MS, various polyphenol classes were identified, such as phenolic acids (benzoic acid and 4-hydroxy-3-methoxybenzoic acid), coumarins (scopoletin), benzenic compounds (catechol and 5-*trans*-butylpyrogallol), and aromatic aldehydes (vanillin) [140]. Other compounds were also identified, such us fatty acids (palmitic, myristic, and linoleic acid), furanic compounds, pyranic compounds, glucopyranose derivatives, and alkyl-phenylketones [141]. Regarding the relative abundance, scopoletin, 5-*trans*-butylpyrogallol, benzoic acid, and linoleic acid were some of the main compounds. Del Cueto et al. [141] also studied the phenolic composition of pruned wood and reported the significant presence of the flavanols catechin, epicatechin, and procyanidin dimers. They also showed a significant amount of chlorogenic acids, aromatic ketone with acetophenone derivatives, and the coumarin scopolin (Table 5).

Recently, researchers have shown interest in the gum exudate obtained from the tree shoots of apricot trees. This fluid/semisolid was composed of a high-molecular-weight polysaccharide with an arabinogalactan structure. More specifically, the polysaccharide contained mainly the monosaccharide xylose, arabinose, and galactose, with a respective ratio of 1:8:8 [142]. This natural polymer has attracted the attention of many researchers in recent years, owing to its wide range of hydrocolloid applications in different industries.

##### Application of Apricot Wood Waste in the Agri-Food Industry

Pruned wood from apricot groves was investigated for its antioxidant capacities and showed high DPPH scavenging activity (1600–3000 mg TE/g), reducing power (300–770 TE/g), and lipid peroxidation inhibition, with a value of 56.2% for its antioxidant activity (AA) [140]. Apricot wood extracts have also been proposed as fungicides, due to the interesting compounds they contain. In fact, due to the presence of polyphenols, known antifungal agents, a study showed the inhibition of *Monilinia laxa*, a brown rot responsible for diseases in stone fruits [141]. Scopoletin and acetophenone derivatives appeared to be the main contributors to the antifungal activity observed. Pruned apricot wood could, therefore, be valorized in farming and integrated perfectly into a circular economy system.

Focusing on gum exudate obtained from the pruned wood of apricot trees, this polysaccharide-based material offers high potential for use as an edible coating or as an emulsifying, binding, and stabilizing agent in the food and pharmaceutical industries [143,144,145]. Jamila et al. [146] showed that the scavenging activity of apricot gum exudate against DPPH radical was higher than that of gums, such as *Salmalia malabarica*, *Acacia arabica*, and *Acacia modesta*. The synthesis of nanoparticles using natural products, such as gums, has attracted a lot of interest; therefore, apricot gum was used to synthetize gold and silver nanoparticles, which demonstrated significant antimicrobial activity against *S. aureus* [147].

As described for plum and cherry wood, toasted chips of apricot wood were also studied in spirit aging. In this sense, several stages of toasting (110, 175, and 200 °C) of apricot wood were applied and the resulting brandy was evaluated for its antioxidant capacity, total polyphenol content, changes in color, and sensory acceptability. Samples of spirits containing apricot wood fragments treated at a lower temperature reached a higher polyphenol content, with the highest antioxidant capacity, while an intermediate temperature promoted a better color intensity. Globally, apricot wood has the potential to be used in brandy production and could be valuable in the beverage industry [144]. A summary of the bioactive compounds isolated from stone fruit wood and their applications can be found in Table 5.

### 3.6. Citrus Fruit Tree

Regarding citrus fruits, oranges (mainly *Citrus sinensis* as “sweet orange” and *Citrus x aurantium* as “sour orange”) represent the most produced fruit, with more than 58% (6.1 MT) of the European production (Figure 7) (European Commission- DG Agri G2, 2020). Smaller citrus fruits, such as mandarins (*Citrus reticulata*), clementines (*Citrus clementina*), or satsumas (*Citrus unshiu*), are also produced in large amounts, representing 27% (2.9 MT) of the production, while lemons (*Citrus limon*) account for 14% (1.5 MT). To a lesser extent, grapefruit (*Citrus paradisi*) and pomelo (*Citrus maxima* sin. *Citrus grandis*) are also cultivated in Europe (1%, 0.1 MT).

Covering over 500,000 hectares in Europe, the cultivation of citrus fruits generates a significant amount of wood residues through the management of trees [148]. One study evaluated the amount of pruned biomass from the main varieties of orange (Navelate, Naveline, Washington Navel, Valencia Late, Salustiana, etc.) and mandarin (Clemenpons, Clemenules, Orogrande, Oronules, Primosol, Owall, Clemenvilla, etc.). Regarding orange cultivars, the Naveline and Washington Navel varieties generate 2.9 tons and 2.4 tons of dried wood biomass/ha, respectively, while the Valencia Late variety produces 4.7 tons of dry biomass/ha, about 48% more than the other varieties [10]. Focusing on smaller citrus fruits, the hybrid cultivars Clemenvilla and the satsuma variety Owal produce the largest amount of wood waste with 7.3 and 6.5 tons of biomass/ha, respectively. While most of the mandarin cultivars generate between 2.8 and 4.4 tons of biomass/ha, some varieties have a lower amount of pruned biomass, with less than 2 tons of biomass/ha, highlighting the variation according to the citrus fruit species and varieties [10]. A recent study of Italian orange orchards reported that annual pruning generates approximately 1.8 tons of wood biomass/ha, which agrees with the Spanish data [149]. Eugenio et al. [150] stated that each year Spain generates 2.25 MT of orange tree pruning residues overall, while González et al. [151] estimate that this Iberian country produces 5 MT of orange wood waste. In this sense, as orange production represents 58% of the citrus production, the pruning from citrus fruit in Spain can be estimated at between 4 and 9 MT per year. Moreover, as Spain accounts for half of the European citrus production, the amount of pruning residues in Europe could be estimated at between 8 and 18 MT per year.

#### 3.6.1. Orange Wood Waste

##### Bioactive Compounds from Orange Wood Waste

Regarding orange tree residues, wood extracts belong to different chemical families. A recent study into pruning biomass from sweet orange (*C. sinensis*) showed the presence of polyphenols and alkaloids [149]. Using LC-MS, the authors showed the main compounds to be the cinnamic acid derivatives caffeic acid, the flavan-3-ol catechin, and an unknown glycosylated compound with a relative abundance at 69.7%, 14.4%, and 15.1%, respectively. Using the Folin–Ciocalteu index, the authors estimated a total polyphenol content of between 35 and 50 GAE/g, depending on the extraction process [149]. Focusing on the composition of wood biomass from bitter orange (*C. aurantium*) trees, a recent study screened a total of 295 metabolites and found 39 compounds using HPLC-Q-TOF-MS [152]. These included three amino acids (asparagine, glutamine, and pipecolic acid), two ribonucleosides ((hydroxy-)adenosine), ten flavanones (derivatives of eriodictyol, hesperitin, hesperidin, and naringenin), fifteen flavones (derivatives of luteolin, diosmetin, acacetin, apigenin, 3′,4′,5′-trimethoxyflavone, cirsimaritin, nobiletin, and tangeretin), two coumarins (xanthotoxol and scopoletin), one alkaloid (M-acetylnorsynephrine-rhamnoside), and six new compounds (citrus A, B, C, H, I, J) as flavonoids, alkaloids, and terpenoids [152] (Table 6).

Wood waste from orange trees has also been subjected to hydrodistillation to produce essential oils. Essential oils from pruned tree shoots of sweet orange trees (*C. sinensis*) were analyzed by GC-MS and exhibited 24 volatile compounds. The main compounds were the monoterpene sabinene, followed by limonene, δ-3-carene, terpinen-4-ol, trans-ocimene, and β-myrcene [153]. Another work, focusing on essential oil from the wood of bitter orange (*C. aurantium*), showed that the major compound was the monoterpene limonene, followed by dimethyl anthranilate, β-fenchol, dodecane, 4-carvomenthenol, ɣ-terpinene, cis-4-thujanol, and thymol [154].

##### Application of Orange Wood Waste in the Agri-Food Industry 

As described for other wood waste, the most common practice for the disposal of these citrus pruning residues is their controlled combustion in the field, allowing ashes to be incorporated into the soil as an amendment [155]. When open burning is forbidden because of possible environmental risks (zones near natural reserves), the wood waste is ground and used as green manuring. However, wood tree biomass is known to be rich in secondary metabolites, which can be valorized in different industrial sectors, especially in the agri-food industry [156,157].

Regarding extracts of sweet orange wood, they showed antioxidant capacity through three different tests. DPPH, FRAP, and BCB (β-carotene bleaching assay) allowed for the measurement of their radical scavenging activity, reducing power, and lipid peroxidation inhibition, respectively. Wood extract obtained from a water autoclave extraction had high DPPH scavenging activity (35.4 mg TE/g) and reducing power (87.3 mg TE/g), while an ethanolic extract exhibited better lipid peroxidation inhibition, with an antioxidant activity (AA) value of 70.3% [149]. Despite orange wood extracts showing good antioxidant properties, as far as we know, no studies are available in the literature regarding their in vitro or in vivo bioactivities or their potential applications. In this sense, further investigation is required and represents an interesting challenge, due to the large amount available and the complexity of orange wood chemical profiles.

The possible application of essential oil derived from orange tree pruning has also been investigated. Eldahishan et al. [153] studied the antibacterial and antifungal activities of *C. sinensis* tree shoot essential oil with a concentration of 12.5 µL against Gram (+) bacteria (*S. aureus*, *S. pyogenes*, and *E. faecalis*), Gram (-) bacteria (*K. pneumoniae*, *E. coli*, and *S. typhimurium*), and fungi (*Trichophyton mentagrophytes*, *Aspergillus fumigatus*, and *C. albicans*). The oil showed a better antibacterial effect against *S. aureus* (responsible for skin and soft tissue infections, i.e., abscesses, furuncles, and cellulitis), *S. pyogenes* (responsible for strep throat), and *S. typhimurium* (causes a protracted bacteremic illness referred to as typhoid fever), as well as antifungal properties against *A. fumigatus* (responsible for severe pulmonary allergic disease) [153]. Furthermore, Okla et al. [154] reported the antibacterial effect of essential oil extracted from bitter orange wood against *Agrobacterium tumefaciens*, *Dickeya solani*, and *Erwinia amylovora*. Overall, essential oils from the hydrodistillation of orange pruning residues showed antibacterial and antifungal properties, allowing it to be valorized for medical purposes and for use in cosmetic products and food additives.

#### 3.6.2. Mandarin Wood Waste

##### Bioactive Compounds from Mandarin Wood Waste

Mandarins (*C. reticulata*) are one the smallest citrus fruits produced in Europe, and research has been performed into the chemical composition of the woody residues generated every year. A study of mandarin tree wood was performed with an acetonic extraction. Six compounds were described: two acridone alkaloids (citruscridone and citrusinine-I), a polyphenol cinnamic acid (valencic acid), two triterpenes limonoid (limonexic acid and limonin), and a coumarate ester (*p*-hydroxyphenyl-ethyl-*p*-coumarate) [158] (Table 6). Mandarin wood is thus composed of a mixture of different chemical compounds that could involve various molecular targets and several signaling pathways.

##### Application of Mandarin Wood Waste in the Agri-Food Industry

Wood extracts from *C. reticulata* and its main compounds were evaluated for their antibacterial and antifungal activities [158]. An acetonic extract showed better antimicrobial activity against four bacteria (*S. aureus*, a methicillin-resistant strain of *S. aureus*, *E. coli*, and *P. aeruginosa*) and antifungal activity against two yeasts (*C. albicans* and *Cryptococcus neoformans*) and a dermatophyte (*Microsporum gypseum*). Among the purified compounds of the acetonic extract, the alkaloid citruscridone and the polyphenol valencic acid inhibited the growth of *M. gypseum*, with MIC values of 200 and 128 mg/L and minimum fungicidal concentration values of 200 mg/L for both compounds. The other compounds (citrusinine-I, limonexic acid, *p*-hydroxyphenyl-ethyl-*p*-coumarate, and limonin) showed no activities up to a dose of 200 mg/L [158]. Overall, mandarin wood waste, rich in polyphenols (flavonoids, phenolic acids, and coumarins), alkaloids (acridone and acridine) and terpenes, could be valorized for medical purposes, in cosmetic products, or as a food additive, thanks to its antibacterial and antifungal properties.

#### 3.6.3. Lemon Wood Waste

##### Bioactive Compounds from Lemon Wood Waste

Lemon (*C. limon*) trees generate a significant amount of wood waste in orchards. However, few studies have focused on the chemical characterization and potential biological activities of the wood or tree shoots of lemon trees. The only available study deals with the valorization of the woody biomass of lemon through essential oils. The essential oils of *C. limon* were isolated from freshly collected tree shoots by hydro-distillation. Using a GC-MS apparatus, 35 compounds were identified that accounted for 99.22% of the essential oils from the pruned wood of lemon trees. Among them, the main compounds were the monoterpene limonene, geranial, neral, neryl acetate, and geranyl acetate [159].

##### Application of Lemon Wood Waste in the Agri-Food Industry

Although the bioactivities of essential oil from wood biomass have not been explored, a lemon leaf oil with a similar composition (i.e., limonene as the main compound, followed by geranial and neral) was tested for its cytotoxic activity against breast carcinoma cells (MCF-7). The lemon essential oil exerted a significant increase in the expression levels of the apoptotic protein caspase-8, as well as a significant decrease in the expression levels of both the anti-apoptotic protein BcL-2 and the proliferative marker Ki-67. In addition, the compound geranial showed a high affinity against caspase-8 and caspase-9, demonstrating the promising cytotoxic effect of lemon oil through enhancing apoptosis [159]. Overall, although further studies are needed, it appears that the wood biomass from lemon tree pruning could be valorized for medicinal purposes, owing to its anticancer properties.

#### 3.6.4. Grapefruit and Pomelo Wood Waste

##### Bioactive Compounds from Grapefruit and Pomelo Wood Waste

Although grapefruit (*C. paradisi*) and pomelo (*C. maxima*) represent less than 1% of citrus production in Europe, some studies have investigated the biomolecules present in their woody waste. A study into grapefruit pruning explored the presence of one of the main triterpenoid limonoids, known as limonin. It was found that increases in limonin levels of up to 0.2 g/kg can be found in grapefruit wood, depending on the season [160]. The chemical profile of pomelo wood residues was also determined by rapid resolution liquid chromatography coupled to a quadrupole time-of-flight mass spectrometry (RRLC-MS/QTOF). The analysis of the methanolic extract showed the presence of 22 compounds, including several polyphenols, such as flavones (vicenin-2, rhiofolin, derivatives of apigenin, and luteolin) and flavanones (eriocitrin, narirutin, naringin, melitidin, and naringenin). The extract also contained several coumarins (meranzin hydrate, meranzin, isomeranzin, marmin, bergapten, imperatorin, and isoimperatorin) and terpene limonoids (limonin, isoobacunoic acid, nomilin, and obacunone) [161] (Table 6).

##### Application of Grapefruit and Pomelo Waste in the Agri-Food Industry

To date, the application of grapefruit and pomelo wood waste has not been investigated. A summary of the bioactive compounds isolated from citrus fruit wood can be found in Table 6.

## 4. Challenges and Perspectives

Millions of tons of wood waste are generated yearly. The most frequent use of this waste is its incorporation into the soil. In the recent years, these residues have been used as biomass for energy production, thus reducing the use of fossil fuels, which is a relevant international goal, decreasing greenhouse gas emissions and contributing to mitigating climate change.

Moreover, wood waste is a rich source of secondary metabolites that are not directly involved in the basic functions of growth, development, and reproduction of the organism, but possess many bioactive properties. These compounds have been widely studied in food and plants, and, to a lesser extent, in wood residues. The last ten years have seen a 7-fold increase in the study of wood waste valorization (Scopus^®^). Olive tree wood extracts have shown many health-related properties, which is a reason why they have been proposed for use in pharmacy and medicine, specifically to treat Alzheimer’s disease and diabetes. Grapevine cane extracts have potent antioxidant and antimicrobial properties, thereby being of use in the food industry as an alternative to SO_2_ in winemaking. Chestnut wood extracts possess antifungal activities that could be beneficial in agriculture. Citrus wood extracts, mainly orange and mandarin, have a composition and flavor that are of use in the cosmetics industry. However, as described in the current review, the technology readiness levels (TRL) of these applications are low. There is still a lack of pilot scale studies demonstrating the viability and profitability of wood waste uses. Therefore, considering all the bioactive properties of both wood extracts and the compounds described in them, along with the beneficial effects on sustainability and the contribution to the circular economy, the use of wood extracts deserves to be more widely studied.

## Figures and Tables

**Figure 1 biomolecules-12-00238-f001:**
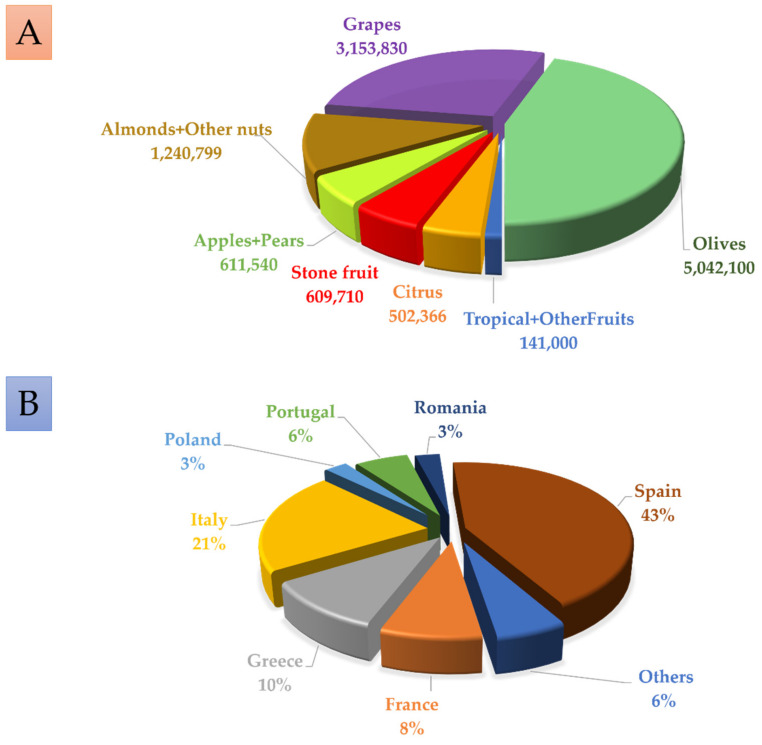
(**A**) Fruit tree area in the EU and (**B**) distribution of fruit trees areas in EU countries (**1B**) [1].

**Figure 2 biomolecules-12-00238-f002:**
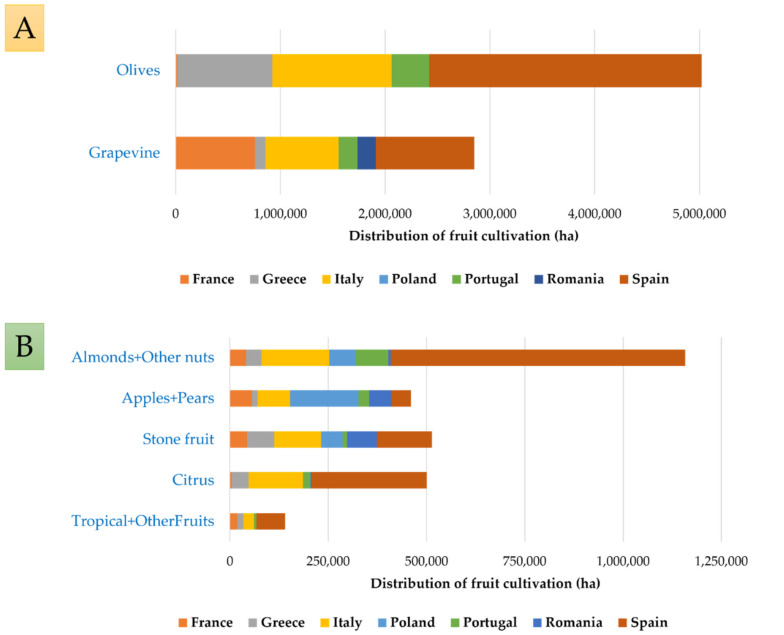
(**A**) Distribution of olive trees and grapevines and, (**B**) nut trees, apple and pear trees, stone fruit trees, citrus fruit trees, and other trees by European countries (ha).

**Figure 3 biomolecules-12-00238-f003:**
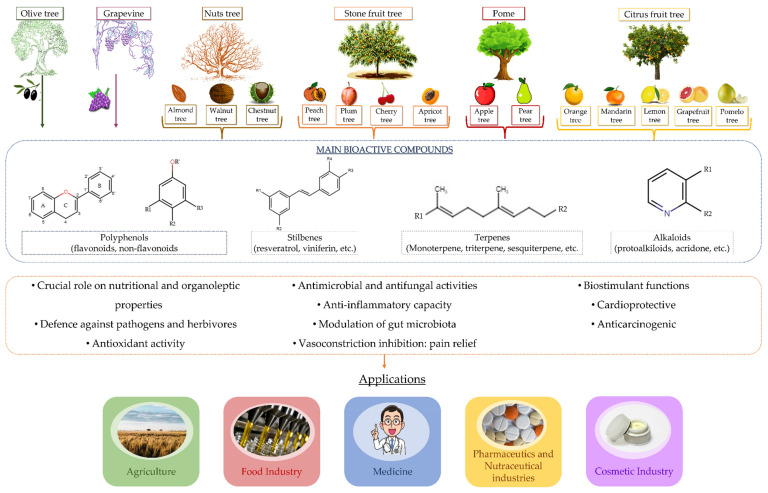
Bioactive compounds present in olive, grapevine, nut, stone fruit, pome fruit, and citrus fruit wood waste, main bioactive properties, and applications reviewed in the current work.

**Figure 4 biomolecules-12-00238-f004:**
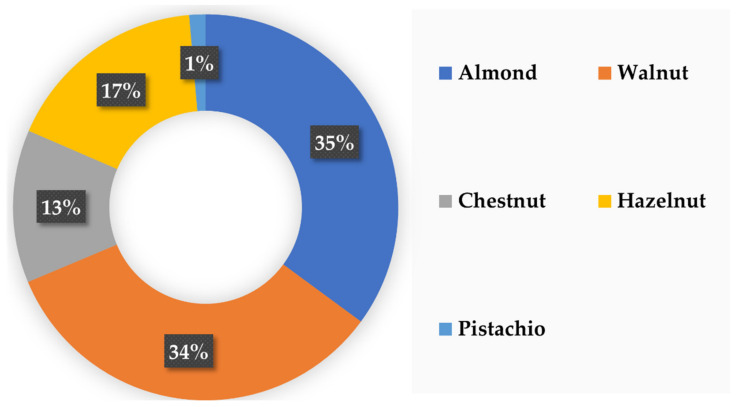
Proportion of nut production in Europe.

**Figure 5 biomolecules-12-00238-f005:**
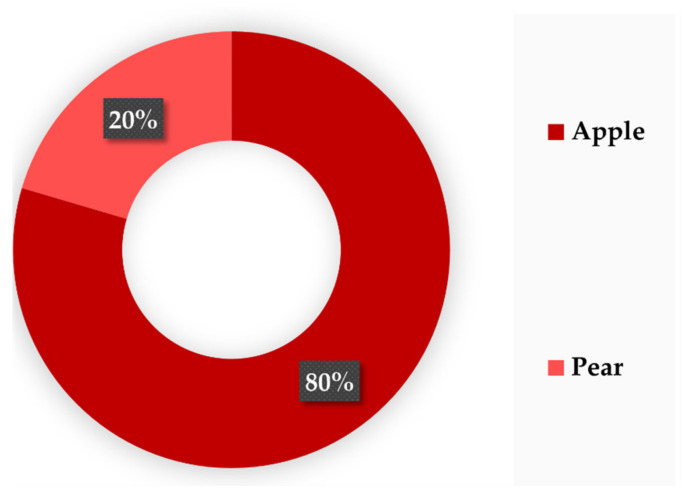
Proportion of pome production in Europe.

**Figure 6 biomolecules-12-00238-f006:**
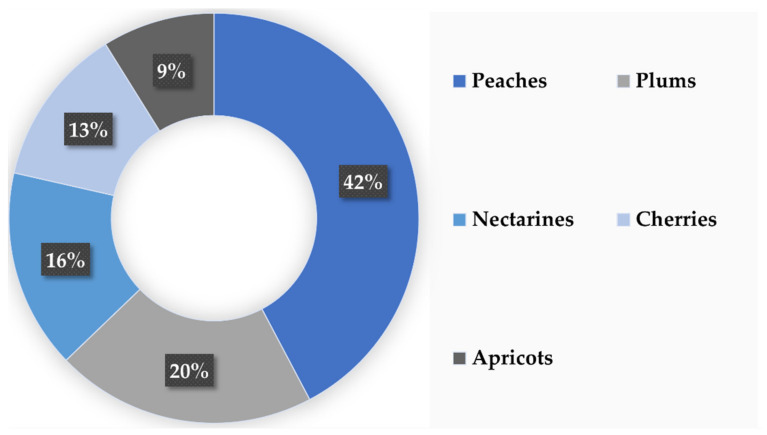
Proportion of stone fruit production in Europe.

**Figure 7 biomolecules-12-00238-f007:**
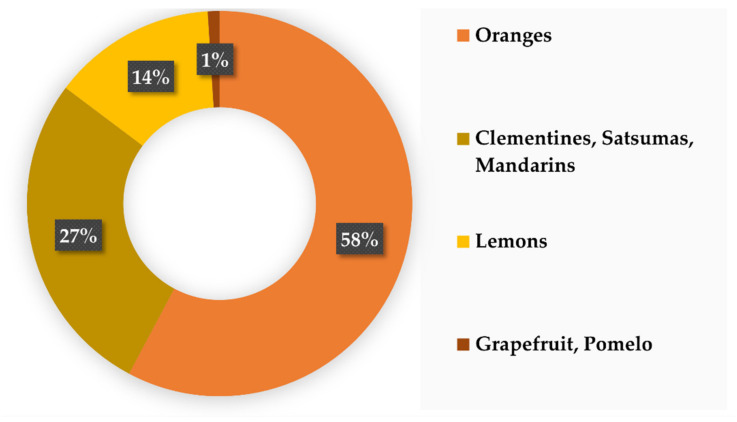
Proportion of citrus fruit production in Europe.

**Table 1 biomolecules-12-00238-t001:** Main bioactive compounds from olive tree wood and their applications. * n.d.: not detected. ** ↑: Increase. Chemical structures by families are shown in Appendix A.

	Family	Compounds (Concentration or Relative Area)	Properties and Applications	References
	Olive wood			
Polyphenol	Benzenoid aromatic	2′-formyl-2,3,4,4′-trimethoxy-1,1′-biphenyl (n.d.–0.72%) 4-(hydroxyacetyl)-1,1′-biphenyl (n.d.–2.1%)	Crucial role in nutritional and organoleptic properties Defense against pathogens and herbivores ↑ Antioxidant activity (188.840 mg TE/kg dw) ↑ Antimicrobial properties (IC50: 80%) ↑ Enzyme inhibition related to Alzheimer’s pathology ↑ Reduction of platelet aggregation in vitro DPPH: 116.5 mg TE/g CUPRAC: 192.91 mg TE/g FRAP: 105.23 mg TE/g	[20,26,34,35,36,37]
	Flavanone	eriodictyol (3000.0–71,030.0 mg/kg dw)		
	Flavone	apigenin-7-glucoside (10.0–15.9) luteolin-7-glucoside (167.0–202.0 mg/kg dw)		
	Phenolic acids	3,4-dihydroxybenzeneacetic acid (n.d.–10.0%) 3,4-dihydroxy-benzeneacetic acid (5.1–5.6%) 3,5-dimethoxy-4-hydroxyphenylacetic acid (n.d.–6.7%) 3-formylbenzoic acid (0.9–1.6) 3-hydroxy-4-methoxybenzoic acid (n.d.–1.22%) 3-hydroxy-butanoic acid (n.d.–0.1%) 4-methoxy-4′,5′-methylenedioxybiphenyl-2-carboxylic acid (n.d.–1.0%) 4-methoxy-4′,5′-methylenedioxybiphenyl-2-carboxylic acid (n.d.–0.9%) 7-deoxyloganic acid (n.d.–1880.0 mg/kg dw) docosanoic acid (0.1–0.2%) eicosanoic acid (0.3–0.5%) hexadecanoic acid (0.7–1.2%) octadecanoic acid (0.1–2.6%) oleic acid (1.0–2.0%)		
	Phenylethanoid	hydroxytyrosol (1500.0–2920.0) tyrosol (1720.0–3640.0)		
	Phenylpropanoids	verbascoside (1338.0–1681.0)		
Alcohols, Aldehydes, and Ketones		1-(2,4,6-trihydroxy-3-methylphenyl)-1-butanone (1.6–1.8) 1-(2,4,6-trihydroxyphenyl)-2-pentanone (n.d.–1.7) 2,6-dimethoxyphenol (0.1–0.7) 2-methoxy-4-vinylphenol (0.4–0.82) 2-methoxyphenol (0.1–0.5) 2-methyl-6-methylene-7-octen-2-ol (n.d.–3.4) 2-methyl-6-methylene-7-octen-2-ol (n.d.–3.8) 3,4-dihydroxyphenyl-2-propanone (n.d.–3.9) 3-methyl-1-(2,4,6-trihydroxy-3-methylphenyl)-1-butanone (n.d.–2.9) 4-((1*E*)-3-hydroxy-1-propenyl)-2-methoxyphenol (5.5–9.4) 4-((1*E*)-3-hydroxy-1-propenyl)-2-methoxyphenol (n.d.–17.4) 4-(4-hydroxyphenyl)-2-butanone (n.d.–1.9) 4,4′-methylenebis[2,6-dimethoxyphenol] (n.d.–0.2) 4-ethoxy-2,5-dimethoxybenzaldehyde (n.d.–5.7) 4-ethoxy-2,5-dimethoxybenzaldehyde (0.5–0.6) 4-hydroxy-2-methylacetophenone (n.d.–0.2) 4-hydroxy-3,5-dimethoxybenzaldehyde (n.d.–0.8) 4-hydroxybenzeneethanol (1.8–3.7) allopurinol (8.1–11.4) benzaldehyde (0.1–0.7) catechol (n.d.–0.5)	↑ Antioxidant activity (Relative Antioxidant Capacity Index: 0.53) DPPH (%): 188.84–26.40 mg TE/g FRAP: 408.80–400.31 mg TE/g	[20]
Amines		1-methyl-*N*-vanillyl-2-phenethanamine (n.d.–10.3%) 2,6-dimethyl-4-pyrimidinamine (n.d.–7.8%) *N*-formyltyramine (n.d.–1.6%)	Crucial role in nutritional and organoleptic properties	[20]
Secoiridoids		(−)-olivil (12,020.0–51,810.0 mg/kg dw) (−)-olivil 4-*O*-β-d-glucopyranoside (5300.0–15,980.0 mg/kg dw) 2″-hydroxyoleuropein (11,920.0–29,610.0 mg/kg dw) isojaspolyoside A (n.d–12,480.0 mg/kg dw) jaspolyanoside (n.d.–13,030.0 mg/kg dw) jaspolyanoside isomer (n.d.–12,130.0 mg/kg dw) jaspolyoside (n.d.–11,220.0 mg/kg dw) jaspolyoside isomer (n.d.–13,400.0 mg/kg dw) ligustroside (13,280.0–21,450.0 mg/kg dw) oleoside 11-methyl ester (n.d.–3760.0 mg/kg dw) oleuropein (22,300.0–58,020.0)	Defense against pathogens and herbivores ↑ Antimicrobial properties (IC50: 80%) ↑ Enzyme inhibition related to Alzheimer’s pathology ↑ Reduction of platelet aggregation in vitro	[20,37]

**Table 2 biomolecules-12-00238-t002:** Main bioactive compounds (mg/kg dry wood) from grapevine wood and their applications. * n.d.: not detected. ** ↑: Increase. Chemical structures by families are shown in Appendix A.

	Family	Compounds (Concentration (mg/kg dw))	Properties and Applications	References
Grapevine wood				
	Anthocyanins	malvidin-3-*O*-(6-*O*-caffeoyl)-glucoside	↑ Antioxidant activity TPC (142 mg GAE g^−1^ dw) ABTS: 0.32–0.70 M Trolox/kg dw Development of nutraceuticals ↑ Antioxidant capacity against the peroxyl radical: ORAC_FL_ 0.40–1.50 M Trolox/g dw ↑ Antimicrobial and antifungal activity IC_50_: 0.97–2.01 mg/mL ↑ Biostimulant action	[49,50,51]
Polyphenols		malvidin-3-*O*-glucoside		
		malvidin-3-*O*-rutinoside		
	Flavonols	kaempferol (n.d.–1800.0)		
		kaempferol-3-*O*-glucoside		
		kaempferol-3-*O*-rutinoside		
		quercetin (0.6–8.2)		
		quercetin-3-*O*-galactoside (6600.0–15,000.0)		
		quercetin-3-*O*-glucoside (200.0–12,6800.0)		
		quercetin-3-*O*-rhabinoside (300.0–2800.0)		
		quercetin-3-*O*-rutinoside (1900.0–41,800.0)		
		myricetin (13.0–63.0)		
		isorhamnetin-3-*O*-(6-*O*-feruloyl)-glucoside		
	Flavan-3-ols and Tannins	catechin (710.0–85,800.0)		
		epicatechin (1000.0–13,300.0)		
		epicatechin gallate (60.0–7800.0)		
		epigallocatechin (1.5–5.4)		
		epigallocatechin–(epi)catechin dimer		
		procyanidin A		
		procyanidin B1 (6200.0–13,730.0)		
		procyanidin B2 (110.0–5100.0)		
		procyanidin B3 (140.0–20,500.0)		
		procyanidin dimer gallate		
	Flavones	luteolin (0.02–0.04)		
	Flavanones	naringin (2.0–27.0)		
		naringenin (11.0–108.0)		
	Flavanonol	astilbin (16.0–1666.0)		
	Hydroxybenzoic acids	gallic acid (70.0–33,000.0)		
		syringic acid (n.d.–32,200.0)		
	Hydroxycinnamic acids	caffeic acid (n.d.–600.0)		
		cinnamic acid (12.0–18.0)		
		*trans*-caftaric acid (40.0–16,100.0)		
		*p*-coumaric acid (150.0–151.0)		
		coutaric acid (n.d.–4.5)		
		ferulic acid (n.d.–2500.0)		
	Stilbenes	ampelopsin A (204.0–220.0)	↑ Antimicrobial properties: essential defense action against infection and injury ↑ Antimicrobial and antifungal activity Protection against pathogens and herbivores	[49,52,53,54]
		ampelopsin F (300.0–360.0)		
		ampelopsin H (20.0–40.0)		
		hopeaphenol (n.d.–1468.0)		
		isohopeaphenol (100.0–120.0)		
		miyabenol C (n.d.–1060.0)		
		pallidol (60.0–80.0)		
		parthenocissin A (200.0–220.0)		
		r2-viniferin (60.0–15,200.0)		
		r-viniferin (n.d.–1116.0)		
		ε-viniferin (47.0–40,600.0)		
		*trans*-piceatannol (4.0–1710.0)		
		*trans*-piceid (36.0–284.0)		
		*trans*-resveratrol (11.0–66,200.0)		
		viniferol E (100.0–140.0)		
		w-viniferin (20.0–40.0)		

**Table 3 biomolecules-12-00238-t003:** Main bioactive compounds from nut tree wood and their applications. * ↑: Increase; ↓: Decrease. Chemical structures by families are shown in Appendix A.

	Family	Compounds (Concentration or Relative Area)	Properties and Applications	References
**Almond wood**				
Polyphenol	Phenolic acid	protocatechuic acid (16.4 g/kg) 4-hydroxybenzoic acid (12.1 g/kg) salicylic acid (11.4 g/kg) vanillic acid (7.9 g/kg) syringic acid (3.6 g/kg) ferulic acid (1.4 g/kg) *p*-coumaric acid (1.4 g/kg)	↑ in vivo learning and memory performance ↑ antibacterial activity ↑ antioxidant capacity: DPPH = 4.5–12.9 µmol TE/g ↓ β-glucosidase enzyme	[84,85,86,87,88]
Organic compound	Cinnamaldehyde	sinapinaldehyde (0.7 g/kg) coniferyl aldehyde (1.4 g/kg)		
Polysaccharide	-	glucan (223 g/kg) xylan (111 g/kg)		
**Walnut wood**				
Polyphenol	Phenolic acid	gallic acid (19.8–98.0 mg/kg) ellagic acid (101.4–393.0 mg/kg) ferulic acid (48.2 mg/kg) chlorogenic acid (1.8–4.9 mg/kg) neochlorogenic acid (1.3–4.5 mg/kg) *p*-coumaric acid (1.9–3.3 mg/kg) protocatechuic acid caffeic acid vanillic acid syringic acid salicylic acid	↑ antioxidant capacity: DPPH = 95–180 mg/L FRAP = 0.9–1.7 µmol AAE/mg BCB = 757–1232 mg/L ↑ antifungal activity ↑ antibacterial activity wine aging: ↑ proanthocyanidins ↓ wine sensory descriptors ↑ anti-inflammatory capacity: ↓ lymphocyte T-cell activation/growth ↓ secretion of interleukin-4,-10,-γ ↓ nitric oxide	[84,89,91,92,93,94,95,96]
	Flavonol	quercetin-3-β-d-glucoside (38.3–150.2 mg/kg) quercetin (0.2 mg/kg) kaempferol (0.1 mg/kg)		
	Flavanols	epicatechin (0.1–0.6 mg/kg) procyanidin B1-B2 (0.1 mg/kg)		
	Aldehydes	Vanillin 4-hydroxybenzaldehyde 3,4-dihydroxybenzaldehyde		
	Cinnamaldehyde	coniferyl aldehyde sinapinaldehyde 4-hydroxycinnamylaldehyde		
Terpene	Sesquiterpene	Curzerene		
Fatty acid	Ester	butyl palmitate isobutyl stearate		
Phenylpropanoïde	Phenylpropene	Elemicin		
Polysaccharide	-	glucan (265 g/kg) xylan (106 g/kg)		
**Chestnut Wood**				
Polyphenol	Ellagitannin	ellagic acid (461.4 g/kg) vescalin (68.5 g/kg) vescalagin (13.0 g/kg) castalin (77.2 g/kg) castalagin (11.8 g/kg) 1-*O*-trihydroxibenzene castalagin punicalagin pedunculagin roburin A–E galloyl glucose valoneic acid dilactone castavaloninic acid	↑ antioxidant capacity: DPPH scavenging = 1234 mg TE/g) FRAP = 2263 mg TE/g ↑ oxidative stability of meat by adding extract in pig diet ↑ antioxidant status and cholesterol metabolism by adding extract in broiler diet ↑ modulation of the gut microbiota ↑ antifungal activity ↑ antibacterial activity ↑ wine aging with high value phenolic compounds ↑ anticarcinoma activity towards breast cancer	[100,101,102,103,104,105,106,107,108]
	Gallotannin	gallic acid (4.5 g/kg)		
Terpene	Triterpene	bartogenic acid bartogenic acid 28-*O*-β-d-glucopyranosyl ester chestnoside A–B		
	Monoterpene	dehydrovomifoliol (2.72 mg/kg) 3-oxoretro-α-ionol (2.39 mg/kg)		
Organic compound	Lactone	β-methyl-γ-octalactone (30.2 mg/kg)		
	Aliphatic aldehyde	nonanal (3.3 mg/kg) 3-octen-1-ol (0.6 mg/kg)		
	Aromatic aldehyde	syringaldehyde (61.4 mg/kg) vanillin (28.1 mg/kg) sinapinaldehyde (19.4 mg/kg) propiovanillone (12.2 mg/kg) butyrovanillone (6.5 mg/kg) *trans*-isoeugenol (7.03 mg/kg) methyl syringate (5.8 mg/kg)		
	Furane	furfural (0.6 mg/kg) 5-methyl-furfural (0.16 mg/kg) 5-hydromethylfurfural (0.39 mg/kg)		

**Table 4 biomolecules-12-00238-t004:** Main bioactive compounds (mg/kg dry wood) from apple tree wood and their applications. * ↑: Increase. Chemical structures by families are shown in Appendix A.

	Family	Compounds (Concentration (mg/kg dw))	Properties and Applications	References
Apple wood				
Polyphenols	Dihydrochalcone	phloridzin (6890.0–8770.0) phloretin (167.0–195.0)		
	Flavanol	(+)-catechin (103.0–301.0) (−)-epicatechin (−)-epicatechin gallate procyanidin B1 procyanidin B2	↑ antioxidant activity DPPH: 28.4 mg Trolox eq/g FRAP: 36.1 mg ascorbic acid eq/g ↑ antimicrobial activity: 68% growth reduction of *Enterococcus faecalis* and *Staphylococcus aureus*	[113,114]
	Flavonol	kaempferol (123.0–219.0) kaempferol-3-*O*-glucoside (1500.0–1960.0) kaempferol-3-*O*-rutinoside (76.9–161.0) quercetin (214.0–229.0) rutin (68.3–98.4)		
		myricetin (1930.0–2590.0)		
	Flavanones	naringin (n.d.–182.0) naringenin (n.d.–31.0)		
	Phenolic acid	4-hydroxyphenilacetic acid (395.0–677.0) caffeic acid chlorogenic acid (n.d.–185.0) cinnamic acid (403.0–815.0) *p*-coumaric acid ferulic acid (40.9–89.0) gallic acid (n.d.–43.1) protocatechuic acid (183.0–269.0) sinapic acid (63.0–113.0) vanillic acid		
	Stilbenes	resveratrol (157.0–320.0)		

**Table 5 biomolecules-12-00238-t005:** Chemical composition, quantitation, major bioactivities, and applications of wood from stone fruit trees. * ↑: Increase; ↓: Decrease. Chemical structures by families are shown in Appendix A.

	Family	Compounds (Concentration or Relative Area)	Properties and Applications	References
**Peach wood**				
Polyphenols	Flavonol	quercetin 3-*O*-β-d-glucopyranoside	↑ antioxidant capacity ↑ anti-lipase activity ↓ acetylcholinesterase ↑ vasorelaxant effect ↓ vasoconstriction inhibition ↑ mushroom growth	[120,127,128]
	Flavanol	catechin, 4′-*O*-methylcatechin		
	Chalcone	4,2′,4′-trihydroxy-6′-methoxychalcone 4′-*O*-β-d-glucopyranoside		
	Phenolic acid	ferulic acid		
	Other	phenyl *O*-β-d-glucopyranoyl-(1→6)-β-d-glucopyranoside		
Terpene	Triterpene	oleanolic acid ursolic acid		
**Plum wood**				
Polyphenol	Flavanol	catechin (0.82 g/kg) epicatechin (0.61 g/kg) dimeric A-type proanthocyanidins: (−)-*ent*-epicatechin-(2α-*O*-7,4α-8)-catechin (2.49 g/kg) (−)-*ent*-epicatechin-(2α-*O*-7,4α-8)-epicatechin (1.72 g/kg) (−)-*ent*-epiafzelechin-(2α-*O*-7,4α-8)-catechin (1.04 g/kg) (+)-epiafzelechin-(2β-*O*-7,4β-8)-epicatechin (1.95 g/kg) (+)-epiafzelechin-(2β-*O*-7,4β-8)-afzelechin (1.31 g/kg) (−)-ent-epiafzelechin-(2α-*O*-7,4α-8)-epiafzelechin)	↑ antimicrobial activity ↑ antibiofilm activity (*Enterobacter* sp.) ↑ wine aging quality improved organoleptic evaluation increased	[129,131]
	Other	annphenone (2.34 g/kg)		
**Cherry wood**				
Polyphenols	Flavanonol	taxifolin (0.09–8.46 g/kg) aromadendrin (0.08–4.54 g/kg) aromadendrin-7-*O*-glucoside (0.08 g/kg)	↑ antimicrobial activity ↑ antibiofilm activity (*Enterobacter* sp.) ↑ antioxidant capacity: DPPH scavenging: 299.9 mmol TE/kg ↑ wine aging transfer of flavonoids to wine stabilization tannins and pigments markers for red-fruit attributes	[101,132,133,134,135,136,137,138,139]
	Flavanone	chrymbrin (0.08–1.85 g/kg) naringenin (0.17–0.41 g/kg) naringenin-7-*O*-glucoside (0.28 g/kg) sakuranetin (0.19 g/kg) sakuranetin-5-*O*-glucoside (0.07 g/kg) eriodictyol (0.09 g/kg) liquiritin		
	Flavanol	catechin (0.32–30.15 g/kg) epicatechin (0.36 g/kg) procyanidin B1 (0.15 g/kg) procyanidin B2 (0.72 g/kg) dimeric B-type proanthocyanidins (3.65 g/kg) trimeric B-type proanthocyanidins (1.25 g/kg)		
	Flavone	chrysin (0.22–0.72 g/kg) tectochrysin (0.08 g/kg) apigenin (0.23 g/kg) dihydrowogonin (0.16 g/kg)		
	Isoflavone	genistein (0.14 g/kg) daidzein		
	Phenolic acid	benzoic acid (0.06 g/kg) ellagic acid (0.27 g/kg) *p*-coumaric acid (26.30 mg/kg)		
	Anthocyanin	pelargonidin 3-*O*-glucoside (10.9%)		
	Coumarin	aesculetine (0.10 g/kg) scopoletin (2.42 mg/kg)		
Terpene	Monoterpene	dehydrovomifoliol (8.39 mg/kg) 3-oxoretro-α-ionol (2.84 mg/kg) 3-oxo-α-ionol (9.17 mg/kg)		
Organic compound	Aromatic aldehyde	vanillin (4.68 mg/kg) isoeugenol (1.31 mg/kg) coniferaldehyde (2.72 mg/kg) sinapinaldehyde (3.56 mg/kg)		
	Aliphatic aldehyde	nonanal (0.28 mg/kg) decanal (0.08 mg/kg)		
**Apricot wood**				
Polyphenol	Phenolic acid	benzoic acid (0.72–4.07%) 4-hydroxy-3-methoxybenzoic acid (0.78–2.23%) chlorogenic acid (2.35–5.55 g/kg) neochlorogenic acid (0.10–0.39 g/kg)	↑ antioxidant capacity DPPH = 3000 mg TE/g FRAP = 300–770 mg TE/g ABTS = 56.2% antioxidant activity ↑ antifungal activity ↑ antimicrobial activity ↑ wine aging antioxidant increased polyphenol increased color intensity increased	[140,141,142,143,144,145,146,147]
	Flavanol	catechin (2.01–2.85 g/kg) epicatechin (0.74–0.75 g/kg) dimeric proanthocyanidins (1.38–4.29 g/kg)		
	Coumarin	scopoletin (3.97–9.87%) scopolin (0.53–0.70 g/kg)		
	Benzenes	catechol (0.73–1.85%) 5-*trans*-butylpyrogallol (4.13%)		
Organic compound	Aromatic aldehyde	vanillin (1.33%)		
	Aromatic ketone	2′,6′-dihydroxy-4′–methoxy acetophenone (0.24–1.72 g/kg) hexoside (0.24–1.72 g/kg)		
Fatty acid		palmitic (1.45–2.25%) myristic (0.41–0.62%), linoleic acid (0.45–9.27%)		
Polymer		xylose arabinose galactose		

**Table 6 biomolecules-12-00238-t006:** Chemical composition, quantitation, major bioactivities, and applications of wood biomasses from citrus trees. * ↑: Increase. Chemical structures by families are shown in Appendix A.

	Family	Compounds (Concentration or Relative Area)	Properties and Applications	References
**Orange wood**				
(*C. sinensis*)				
Polyphenol	Phenolic acid	caffeic acid (69.7%)	↑ antioxidant capacity: DPPH scavenging = 35.4 mg Trolox eq/g) FRAP = 87.3 mg Trolox eq /g BCB = 70.3% antioxidant activity ↑ antibacterial activity (*S. aureus*, *S. pyogenes*, and *S. typhimurium*) ↑ antifungal activity (*A. fumigatu*)	[149,152,159]
	Flavanol	catechin (14.4%)		
	Lignan	Unknown—*m*/*z* 384 (1.0%)		
Alkaloid	-	Unknown—*m*/*z* 323 (7.4%)		
Essential oil	Monoterpene	sabinene (33.0%) limonene (18.7%) δ-3-carene (9.4%) terpinen-4-ol (6.2%) *trans*-ocimene (6.1%) β-myrcene (4.4%)		
(*C. aurantium*)				
Polyphenol	Flavanone	Eriodictyol eriodictyol-*O*-glucoside hesperitin hesperitin-*O*-rhamnoside hesperitin-*O*-glucoside hesperidin neohesperidin naringenin-*O*-arabinoside-*O*-rhamnoside-*O*-arabinoside 4′-methoxy-flavanone-*O*-rhamnoside		
	Flavone	luteolin-*O*-glucoside luteolin-*O*-glucoside-*O*-rhamnoside-*O*-glucoside diosmetin-*O*-glucoside diosmetin-*O*-glucoside-*O*-rhamnoside acacetin-*O*-glucuronic acid-*O*-rhamnoside apigenin apigenin-*O*-glucoside-*O*-rhamnoside apigenin-*O*-glucuronic acid-*O*-arabinoise-*O*-rhamnoside 3′,4′,5′-trimethoxyflavone 3′,4′,5′-trimethoxyflavone-*O*-rhamnoside cirsimaritin-*O*-arabinose cirsimaritin-*O*-arabinose-*O*-rhamnoside-*O*-rhamnoside cirsimaritin-*O*-rhamnoside-*O*-arabinose-*O*-arabinose nobiletin tangeretin		
	Coumarin	Xanthotoxol Scopoletin		
Alkaloid	Proto-alkaloid	*m*-acetylnorsynephrine-rhamnoside		
Amino acid		Asparagine glutamine pipecolic acid		
Ribonucleoside		hydroxy-adenosine adenosine		
Essential oil	Monoterpene	limonene (38.1%) β-fenchol (6.8%) 4-carvomenthenol (4.2%) ɣ-terpinene (3.6%) *cis*-4-thujanol (3.49%) thymol (3.3%) linalool (2.9%) 6,7-dihydrogeraniol (2.2%)	↑ antibacterial activity (*A. tumefaciens*, *D. solani*, and *E. amylovora*)	[154]
	Sesquiterpene	valencene (3.3%)		
	Alkane	dodecane (5.3%) undecane (2.1%)		
	Other	dimethyl anthranilate (8.1%)		
**Mandarin wood**				
Polyphenol	Phenolic acid	valencic acid (0.21 mg/g)	↑ antibacterial activity (*S. aureus*, methicillin-resistant strain of *S. aureus*, *E. coli*, and *P. aeruginosa*) ↑ antifungal activity (*C. albicans*, *C. neoformans*, and *M. gypseum*)	[158]
Alkaloid	Acridone	citruscridone (0.15 mg/g) citrusinine-I (0.08 mg/g)		
Terpene	Limonoid	limonexic acid (0.39 mg/g) limonin (0.44 mg/g)		
**Lemon wood**				
Essential oil	Monoterpenes	limonene (55.2%) (*E*)-β-ocimene (1.6%) β-myrcene (1.5%) β-pinene (1.5%)	↑ expression levels of apoptotic protein caspase-8 ↓ expression levels of anti-apoptotic protein BcL-2 and proliferative marker Ki-67. ↑ cytotoxic effect	[159]
	Sesquiterpenes	*trans*-caryophyllene (2,2%)		
	Oxygenated	geranial (7.9%) neral (6.1%) eryl acetate (5.6%) geranyl acetate (5.1%) citronellal (1.7%) nerol (1.4%) *trans*-geraniol (1.1%)		
**Pomelo wood**				
Polyphenol	Flavanone	naringenin (5.2 g/kg) eriocitrin narirutin naringin melitidin		[161]
	Flavone	rhiofolin (0.1 g/kg) vicenin-2 apigenin-6/8-C-glucoside-*O*-arabinoside apigenin-6/8-C-glucoside-*O*-rhamnoside apigenin-8-C-glucoside luteolin-7-*O*-rutinoside		
	Coumarins	meranzin hydrate meranzin isomeranzin marmin bergapten imperatorin isoimperatorin (0.03 g/kg)		
Terpene	Limonoid	limonin isoobacunoic acid nomilin obacunone		
**Grapefruit wood**				
Terpene	Limonoid	limonin (0.2 g/kg)		[160]

## Data Availability

Not applicable.

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
