# Peer review of "Wood Waste from Fruit Trees: Biomolecules and Their Applications in Agri-Food Industry"

_biomolecules, 2022, doi:10.3390/biom12020238_

Round 1

Reviewer 1 Report

This review presents the current state of chemical characterization of wood wastes from fruit trees with the highest demand in Europe and the functional activities for their extracts. The document is well organized with clear writing. But it is suggested to consider stating the main objective, findings, and perspective of the review in the Abstract section. Because the objective seems to be to use the woods wastes for obtaining bioactive compounds which can be useful in different industries, besides biorefinery, it is important to improve the discussion about it, perhaps by including the uses of isolated compounds in each area of ​​interest, i.e. some compounds found in woods are already used as pharmaceuticals, but they have been extracted from different plant sources. Also, it is important to include a comparison of the yielding achieved from woods extracts to those obtained from different sources, even those employed at a trading level.

It is recommended to revise the format and style requested by the journal, there are different forms used in the document for citing references. There are some abbreviations missing or defined after their first appearance in the text. The recommendation is to consider some revisions before accepting this manuscript.

Some specific comments are listed below:

Abstract section. In lines 26-28, it is stated that this type of woods may be used in different applications from agricultural to medical or cosmetic industries, nevertheless, most of the discussion in the document is focused on the evaluation of in vitro activities (antimicrobial, antioxidant, etc.) without any discussion about the current state of their usage in these areas of application.

Line 19. Define EU in these lines.

Line 127, 412, 421, 422, 433. Homogenize the format for units, there are several forms used in the document: t/ha; tons; tons per hectare per year or t/ha/year; tons/ha/year.

Lines 185, 411, 618. Revise the spelling for scientific names.

Lines 220, 224, 363, 420, 430, 488-489, 515, 539, 547, 549, 553,556, 676, 677, 710, 804, 812,  883, 884, 936, 944. revise the format required for citing references and add the citation numbering closer to the author’s name in the text.

Line 254,262, 296, 932. Use italics for in vitro or in vivo terms

Line 232. Define TE in this line instead of line 272.

Line 361. What do the authors mean by bioactive carbon?

Line 424. Homogenize the use of a comma or period for separating the thousands.

Lines 439, 467. It is not clear why using sections 3.3.1.a and 3.3.1.b without defining section 3.3.1. The same occurs in further subsections.

Lines 585-587. Does this mean that both fruits represent the total production and consumption for the pome genus? What is the relevance of Figure 5? The information in the figure is already stated in the text.

Line 590. Does this data occur in the whole EU?

Table 5. Revise the alignment for the compound 4,2',4'-trihydroxy-6'-methoxychalcone 4'-O-β-D-glucopyranoside in the peach wood sample.

Line 864. Revise the spelling in this line.

Line 898. Revise the format for these units and homogenize in the document.

Line 1080. Complete the information for this reference.

Author Response

#Reviewer 1:

  • This review presents the current state of chemical characterization of wood wastes from fruit trees with the highest demand in Europe and the functional activities for their extracts. The document is well organized with clear writing. But it is suggested to consider stating the main objective, findings, and perspective of the review in the Abstract section. Because the objective seems to be to use the woods wastes for obtaining bioactive compounds which can be useful in different industries, besides biorefinery, it is important to improve the discussion about it, perhaps by including the uses of isolated compounds in each area of ​​interest, i.e. some compounds found in woods are already used as pharmaceuticals, but they have been extracted from different plant sources. Also, it is important to include a comparison of the yielding achieved from woods extracts to those obtained from different sources, even those employed at a trading level.

It is recommended to revise the format and style requested by the journal, there are different forms used in the document for citing references. There are some abbreviations missing or defined after their first appearance in the text. The recommendation is to consider some revisions before accepting this manuscript.

First, the authors would like to thank the reviewer for revising the manuscript and the motivating comments. The suggestions have been useful for the improvement of the manuscript. The response to each comment can be found below and changes in the manuscript have been expressed with track changes and underlined in yellow in the case of references.

The abstract has been modified, including more information about findings, objectives, etc. as suggested by Reviewer 1.

In our opinion, to include the bioactivity described for the compounds studied in this review, when they can be found in other sources, would increase the complexity and extension of the manuscript.

All the manuscript has been reviewed to modify format mistakes.

  Some specific comments are listed below:

  • Abstract section. In lines 26-28, it is stated that this type of woods may be used in different applications from agricultural to medical or cosmetic industries, nevertheless, most of the discussion in the document is focused on the evaluation of in vitro activities (antimicrobial, antioxidant, etc.) without any discussion about the current state of their usage in these areas of application.

The abstract has been modified to clarify the goal of the work and some applications have been included. Regarding the current state of the applications, a paragraph has been included in the Challenges and Perspectives Section.

  • Line 19. Define EU in these lines.

It has been done.

  • Line 127, 412, 421, 422, 433. Homogenize the format for units, there are several forms used in the document: t/ha; tons; tons per hectare per year or t/ha/year; tons/ha/year.

The manuscript has been reviewed and units have been homogenized to tons/ha and tons/ha per year.

  • Lines 185, 411, 618. Revise the spelling for scientific names.

Scientific names have been reviewed.

  • Lines 220, 224, 363, 420, 430, 488-489, 515, 539, 547, 549, 553,556, 676, 677, 710, 804, 812, 883, 884, 936, 944. revise the format required for citing references and add the citation numbering closer to the author’s name in the text.

The manuscript has been reviewed and references have been located close to the author's name according to reviewer’s recommendation. In addition, references 33 and 57 have been added. Reference changes have been marked in yellow

  • Line 254,262, 296, 932. Use italics for in vitro or in vivo terms

It has been corrected all through the manuscript.

  • Line 232. Define TE in this line instead of line 272.

TE is Trolox equivalent. It has been corrected in the new version of the manuscript.

  • Line 361. What do the authors mean by bioactive carbon?

Please kindly notice that “bioactive carbon” refers to the carbon obtained after controlled burning of the wood and that it has proven to be rich in some compounds such as minerals, amino acids, organic acids, polysaccharides, and polyelectrolytes in some cases, its composition varying according to the raw material composition and the characteristics of the burning process. This bioactive carbon has been commonly employed as a biostimulant in the field itself to increase the concentration of these compounds in plants.

  • Line 424. Homogenize the use of a comma or period for separating the thousands.

It has been corrected throughout the manuscript.

  • Lines 439, 467. It is not clear why using sections 3.3.1.a and 3.3.1.b without defining section 3.3.1. The same occurs in further subsections.

New subsections have been added to clarify the structure of the manuscript.

  • Lines 585-587. Does this mean that both fruits represent the total production and consumption for the pome genus? What is the relevance of Figure 5? The information in the figure is already stated in the text.

Yes, the consumption of the remaining pome genus fruits is so small that Eurostat database does not include those data.

The authors considered that Figure 5 helps to clarify this idea and it is in line with the other figures included for nuts (Figure 4), stone fruits (Figure 6) and citrus fruits (Figure 7).

  • Line 590. Does this data occur in the whole EU?

Yes, this information is obtained from European statistical data.

  • Table 5. Revise the alignment for the compound 4,2',4'-trihydroxy-6'-methoxychalcone 4'-O-β-D-glucopyranoside in the peach wood sample.

It has been corrected.

  • Line 864. Revise the spelling in this line.

It has been corrected.

  • Line 898. Revise the format for these units and homogenize in the document.

These units have been modified to GAE and homogenised throughout the document.

  • Line 1080. Complete the information for this reference.

It has been completed.

Reviewer 2 Report

Dear Authors,

your manuscript is very interesting and well written. Substantive analysis well ordered.

But there is a drawback that needs to be corrected. There is no scientific aim of the study. Please precise what do you want to prove or disporove? Why such an analysis is needed? And further in the conclusion summarise what new konwledge did you create.

Please add some mthodical data. Where did you search for articles and other sources? How many of them did you find? How they have been selected for the analysis? What were the selection criteria? Etc.

The main text have to be extended by these inforamations.

And also include these information is short sentences in the abstract.

Author Response

#Reviewer 2:

Dear Authors, your manuscript is very interesting and well written. Substantive analysis well ordered. But there is a drawback that needs to be corrected. There is no scientific aim of the study. Please precise what do you want to prove or disporove? Why such an analysis is needed? And further in the conclusion summarise what new konwledge did you create. Please add some mthodical data. Where did you search for articles and other sources? How many of them did you find? How they have been selected for the analysis? What were the selection criteria? Etc. The main text have to be extended by these inforamations. And also include these information is short sentences in the abstract.

First, the authors of the paper would like to thank the reviewer for revising the manuscript and the motivating comments. We have modified the abstract to focus on the relevance and importance of the topic.

The scientific aim of the work is to demonstrate the potential of wood waste as a source of bioactive compounds, with numerous applications. We have tried to clear this goal in the manuscript and in the abstract according to the reviewer’s recommendation.

Regarding methodology, we would like to remark the difficulty of finding references for some of the manuscript sections. Thus, it was not possible to do a systematic review. The methodology used was as follows: we used both the Scopus®  and the ScienceDirect® databases as  the  sources  for  extracting  publications.  The following search query was constructed and applied: 

TITLE-ABS-KEY “wood waste” AND “circular economy”

TITLE-ABS-KEY “olive/pome fruit/pears/apple/stone fruit/nut/citrus fruit” AND “wood” AND “waste”

TITLE-ABS-KEY “olive/pome fruit/pears/apple/stone fruit/nut/citrus fruit” AND “pruned” AND “waste”

TITLE-ABS-KEY “olive/pome fruit/pears/apple/stone fruit/nut/citrus fruit” AND “wood” AND “composition”

TITLE-ABS-KEY “grapevine/canes” AND “wood” AND “composition”

TITLE-ABS-KEY “olive/pome fruit/pears/apple/stone fruit/nut/citrus fruit” AND “tree” AND “wood” AND “bioactive compounds/polyphenols/anthocyanins”

TITLE-ABS-KEY “grapevine wood/canes” AND “bioactive compounds/polyphenols/anthocyanins”

TITLE-ABS-KEY  “agroindustry applications”  AND  “fruit tree wood”

TITLE-ABS-KEY “bioactive compounds” AND “fruit tree wood” AND “applications”

Between 1999  and  2021, a total number of  1,106 published  papers  were  obtained. However, some of these topics have only five researchers, and many of the applications evaluated for the fruit tree wood are based on their burning or use as compost. For this reason, these articles were reduced by a considerable number. The selection was performed depending on the relevance of the content, the novelty of the applications tested and their possible implementation. We think this methodology should not be included in the manuscript since it was not designed as a Systematic Review.

Reviewer 3 Report

The review entitled: Wood Waste from Fruit Trees: Biomolecules and Their Applications in Agri-Food Industry  is related to bioactive properties of wood extracts. The paper is dived into four section and close with literature data.

The first part is dedicated to Abundance and importance of fruit trees in Europe  which is very well documented and argued for the subject of the paper.

The second part, Contribution of wood waste to a circular economy and sustainable bioeconomy,  described  the valorisation of biomass wastes and residues into valuable products.

The chapter 3- Main fruit trees in Europe: bioactive compounds from wood and applications- is the most extensive and is subdivided in six subchapters.  Each subchapter is subdivided and comprises between two and eight parts. All the data are illustrated by five figures and six tables which make the paper friendly reader either for people generally interested in the subject not only for the specialists.

In the last part, Challenges and perspectives, the authors state a very pertinent conclusion, highlighting the fact that wood residues should be studied much more, taking into account all the bioactive properties of wood extracts, along with the beneficial effects on sustainability and contribution to the circular economy.

The review have 160 literature titles correlated with the subject of the work.

Finally two observations:

Line 19: „ ...EU estimates approximately 25 and 2 million tons…”  I would suggest changing the text like this  „ ...EU estimates approximately 2 and 25 million tons…” 

Line 411: Pistacia vera- please use italic form  Pistacia vera

In other words, congratulations to the authors. I find this review very useful !

Author Response

#Reviewer 3:

The review entitled: Wood Waste from Fruit Trees: Biomolecules and Their Applications in Agri-Food Industry  is related to bioactive properties of wood extracts. The paper is dived into four section and close with literature data.

The first part is dedicated to Abundance and importance of fruit trees in Europe  which is very well documented and argued for the subject of the paper.

The second part, Contribution of wood waste to a circular economy and sustainable bioeconomy,  described  the valorisation of biomass wastes and residues into valuable products.

The chapter 3- Main fruit trees in Europe: bioactive compounds from wood and applications- is the most extensive and is subdivided in six subchapters.  Each subchapter is subdivided and comprises between two and eight parts. All the data are illustrated by five figures and six tables which make the paper friendly reader either for people generally interested in the subject not only for the specialists.

In the last part, Challenges and perspectives, the authors state a very pertinent conclusion, highlighting the fact that wood residues should be studied much more, taking into account all the bioactive properties of wood extracts, along with the beneficial effects on sustainability and contribution to the circular economy.

The review have 160 literature titles correlated with the subject of the work.

First, the authors of the paper would like to thank the reviewer for revising the manuscript and the motivating comments.

Finally two observations:

Line 19: „ ...EU estimates approximately 25 and 2 million tons…”  I would suggest changing the text like this  „ ...EU estimates approximately 2 and 25 million tons…”

It has been changed

 Line 411: Pistacia vera- please use italic form  Pistacia vera

It has been corrected

Reviewer 4 Report

The review reflects an idea of biorefining the woody biomass of the lowest quality. The pruning woody residues of fruit tree species are presented as a possible source of bioactive low-molecular compounds. These compounds are also known as wood extractives. Especially those of phenolic character exhibit many beneficial properties, even for human health. The review is comprehensive and well written. A complete review on the fruit production in Europe and the amounts of corresponding pruning residues produced in the orchards is given by the paper. These woody residues are produced mainly by annual pruning and also by renewal of orchards in a smaller extent. Annual amount of pruned wood is estimated for the investigated tree species. All main secondary metabolites with their properties and possible applications are provided for the wood of selected tree species of olives, nuts, apples and pears, stone fruits, and citruses as well as for grapevines shoots. The methodology for qualitative and quantitative evaluation of wood extractives and their bioactive properties is correctly summarized. The manuscript is supported with the correct graphical and tabelaric material that help a reader to follow the story easily. However, I suggest that all the tables showing main compounds, their properties and possible applications are equipped also with chemical structures of these compounds. They would be very useful for a reader/researcher from the field. The quality/resolution of figures needs to be improved. The title of the paper is clear and informative. Abstract is correctly written. All the relevant literature is reviewed and cited. The review is placed in the context in a clear and exact way. Style of writing is clear and adequate to me. Terminology used is correct but does not completely correspond to one that is used by the wood science community. However, when reading the paper three drawbacks arises quickly. As first. The wood of younger parts of trees in my opinion cannot be considered as a relevant source of bioactive phenolic compounds. Maybe for sugars but not for phenolic extractives. Literature data on this exists. The review itself with the numbers mentioned confirms that. Secondly. The review does not report what is happening at higher trl’s? Economic justification of such extractions? After all, biomolecules with beneficial properties are already being extracted out from the residual biomass of trees and are commercially available as e.g. supplements for feed and even as dietary supplements. Thirdly, the idea of valorisation of woody biomass of the lowest quality and integration of these side-streams of the industry to a system of circular economy is already well developed. Lack of novelty is obvious here. Despite mentioned, the paper is well structured, authors summarized the existing literature in a clear and correct way. Therefore, I recommend the editorial board to accept the manuscript for publication after minor revisions. Some additional comments are listed below.

P3, 75-76. This statement should be supported with the numbers for other countries having a role of fruit suppliers, e.g. US, South Africa etc. Please revise.

P4, L125-127. You are writing here about wood? Are here meant only woody tissues? Including bark? Above you are mentioning branches as the pruning residues. Chemical composition of extractives from bark or from wood can differ significantly. Revise, make this part clearer. 

P6, 181-183. The resolution of the scheme on figure 3 has to be improved. The scheme is of bad quality, is hard to read it.

P6, L182. Bioactive compounds present at olive, grapevine, nuts, stone fruits, pome fruits, and citrus fruits wood waste? Which bioactive compounds? Only groups of certain phenolics are presented with the fig.6. Please be more exact. Afterall this is not brochure, it is scientific paper.

P7, L201. Olive trees? What do you mean whit this? Bark, wood, sapwood, heartwood, branch wood, leaves, roots, bark of roots? Which category of tree tissue? Please revise, make it clear. 

P7, L243-312. Applications from olive wood waste? You are summarizing the papers that present results of a lab scale research activities. Many of which were done in vitro. Only conventional techniques for extraction (Soxhlet, sonication) are mentioned.

P13, L411. Pistacia vera. Typo. Italic.

P14, L413-414. A comment on figure 4. Instead using different shades of green, I suggest using different colors. It is hard to distinguish some of the green colors on the graph. The graph is hard to read.

P14, L419. The availability of wood trunk (also known as heartwood). The terminology is not familiar to me. As proposed by IAWA, heartwood is defined as "the inner layers of the wood, which, in the growing tree, have ceased to contain living cells, and in which the reserve materials (e.g., starch) have been removed or converted into heartwood substance" (IAWA 1964, p. 32). So called wood trunk contains also sapwood, these tissues have different physiological role than heartwood. They provide conductive function. The amount of phenolic compounds that can be extracted from sapwood and heartwood differ significantly. Anyhow, revise and make these things clear.

P15, L468-473. These quantities are really low. Mg per kg of material? These numbers demonstrate the pruned wood as a poor source of bioactive molecules.

P25, L829-830. This is too general. Populistic. Polyphenols are actually weak antifungal agents if comparted to synthetic fungicides. They only partially inhibit the growth of fungi. Wood decaying fungi are able to detoxicate and destroy natural phenolic compounds.

P29, L923-921. It is true that certain parts of trees are rich sources of low-molecular weight bioactive molecules, but branchwood is not one of these tissues. This was already scientifically confirmed. Your review somehow confirms that, the reported amounts are really low.

P33, L1025-1026.  … but not in wood residues? Not true. Just for an example, look at the publications of e.g. Oleson et al. 2016 (Extractives in Douglas-fir forestry residue and considerations for biofuel production) and Vek et al. 2021 (In vitro inhibition of extractives from knotwood of Scots pine (Pinus sylvestris) and black pine (Pinus nigra) on growth of Schizophyllum commune, Trametes versicolor, Gloeophyllum trabeum and Fibroporia vaillantii). First attempts of utilizing the wood residues as a source of valuable phytochemicals are actually more than 15 years old, or even more. The idea is relatively old. Please revise the sentence.

P33, L1036-1038. I agree. This research field is one of the most "active" in last years, and receives a lot of scientific attention. 

P33, L1052. The list of references, and figures and tables have to be designed and organized according to the journal guidelines.

Author Response

I attach the answer to Reviewer 4
